# Research on the Dynamic Evolution of the Landscape Pattern in the Urban Fringe Area of Wuhan from 2000 to 2020

Yan Long [1], Shiqi Luo [1], Xi Liu [1], Tianyue Luo [2] and Xuejun Liu [2,3,*]

1    School of Urban Construction, Wuhan University of Science and Technology, Wuhan 430081, China
2    School of Urban Design, Wuhan University, Wuhan 430072, China
3    Research Center for Digital City, Wuhan University, Wuhan 430072, China
*    Correspondence: xjliu@whu.edu.cn

**Abstract:** The urban fringe area is a discontinuous spatial phenomenon that refers to the urban-rural interlacing zone which is undergoing urbanization on the fringe of the core built-up area of a large city after the emergence of industrialization. Dynamic, ambiguous, and complex interlacing of various types of lands make urban planners and managers fuzzy about the spatial scope of the urban fringe and it is difficult to control its evolution patterns scientifically. Based on remote sensing data from 2000 to 2020, the range of Wuhan's urban fringe was extracted from the surface impermeability ratio mutation points, landscape flocculation, and population density. On this basis, the dynamic evolution characteristics of land-use and landscape patterns in the urban fringe area of Wuhan City were analyzed by using dynamic change and landscape pattern index analysis. The results show that: Wuhan City shows a clear "urban core area-urban fringe area-rural hinterland" circle structure, and the urban fringe area continuously extends to the rural hinterland. Moreover, most of the rural hinterland, in the process of moving to the urban core area, has gone through the process of the urban fringe. By comparison with other cities, it is found that the expansion of large cities is generally influenced by policies, topography, and traffic arteries, and gradually shifts from expansion to infill, with the urban core of Wuhan continuously extending and the urban fringe rapidly expanding from 2000 to 2010, and gradually entering a stable development state from 2010 to 2020. The future urban construction of Wuhan should pay attention to the influences of these characteristics on the implementation of urban territorial spatial planning.

**Keywords:** urban fringe area; landscape pattern; dynamic evolution; Wuhan City

## 1. Introduction

With the tide of global urbanization, the spatial distribution of cities shows a changing trend. As a derivative of highly urbanized developed areas, the morphological characteristics, expansion patterns and ecological service functions provided by various types of landscape patches for the central urban areas are the focus of many scholars' research [1–3]. The continuous development of China's economy and the continuous expansion of urban scale have also had a large impact on the morphology, function, and landscape pattern of urban fringe areas. Scholars at home and abroad have studied urban fringe areas mostly from a multidimensional perspective, combining the backgrounds of geography, ecology, sociology, and other disciplines, and have experienced a process from theory to empirical evidence.

Since the 1980s, scholars have started to research numerous problems of urban fringe zones. From the definition and extension of the concept to the division of its scope then to the landscape ecological services provided by the urban fringe zone to the central urban area of the city, and to the study of the coupling relationship between the dynamic changes of the urban fringe zone and the spatial expansion of the city. Among them, Gu Chaolin [4] gave a systematic overview of the urban fringe zone, arguing that the urban

fringe zone has natural and social attributes, is the most effectively studied area of the urban–rural continuum, and is a reflection of urban expansion on agricultural land. Cheng Liansheng [5] used information entropy to determine the urban fringe zone of Beijing and analyzed the trend and rate of expansion. Zhou Xiaochi [6] extracted the urban fringe zone in Xi'an from three aspects of social activities, physical space, and landscape pattern, which greatly improves the accuracy of urban fringe zone range extraction, Dai Junjie [7] based on the mutation characteristics of the urban fringe area, the range of the urban fringe area in Jiangyin City was extracted by identifying multivariate data mutation point groups through wavelet transform detection, which is more complete and objective than the classical method. Since the boundaries of urban fringe areas change with urban spatial expansion and urban–rural relations, it is difficult to determine the boundaries of this dynamic region [8].

Landscape pattern refers to the long-term composition of landscape ecological processes, forming factors, and the spatial distribution of landscape patches of different shapes and sizes [8]. Studying the dynamic change of landscape pattern can reveal the intrinsic relationship between the structural characteristics of landscape elements and the interaction of ecological processes, and provide ecological service function for the protection of landscape ecology. The research on landscape patterns first originated in foreign countries. In 1939, the German geographer Troll first proposed the concept of "landscape ecology" and studied the overall structure and function of the landscape. Between 1981 and 1983, Forman proposed the "patch-corridor-matrix. From 1981 to 1983, Forman proposed the "patch-corridor-matrix" model, which laid the foundation for the study of landscape ecology. Since then, research on landscape patterns has gained sufficient space for development in foreign countries, and occupied a rather important position in the field of landscape ecology research [9]. In 1999, R. Hietala-Koivu used landscape index analysis to study changes in agricultural landscape patterns in the Yläne region of southwestern Finland over a 39-year period. In 2000, Griffith used landscape pattern analysis software to summarize 27 landscape pattern indicators and characteristics of the Kansas Land Cover Database, USA, at three different spatial resolutions [10]. A series of studies have also been conducted in China [11–13]. The evolution process of land-use landscape pattern and its driving forces in Xi'an city from 1995 to 2013 are analyzed, and the overall landscape diversity index, fragmentation, and homogeneity are decreasing, and the degree of landscape heterogeneity is decreasing, and also the main influencing factors are economy and population [14]. Tian Yu [15] analyzed the dynamic evolution process of land-use landscape pattern in Yubei District, Chongqing from 2000–2015, which showed that the degree of landscape fragmentation increased and landscape diversity increased. Hu Fangwen [16] studied the landscape pattern of park green space in urban areas of Shanghang County and found that from 2009–2018, the park green space aggregation index decreased, the degree of fragmentation increased, and the number and simple shape of landscape patches are not conducive to the establishment and maintenance of urban biodiversity. Different types of landscape patches are the main components of the landscape pattern, maintaining the diversity of landscape patches, exploring the evolution pattern of landscape patches, and analyzing the degree of fragmentation of landscape patches can reflect the relationship between landscape pattern and ecological processes, i.e., landscape ecological services.

The urban fringe is a territory where the built-up area of the city and the vast surrounding agricultural land merge and gradually change [17]. Its spatial continuity and gradual change of land feature vectors make it an independent territorial spatial unit between the city and the countryside, providing an important landscape ecological function for urban development. Moreover, the large landscape patches in the fringe area are the best carrier to carry this. Therefore, it is especially important to study the landscape pattern of urban fringe areas. In recent years, some scholars have analyzed the evolution of landscape patterns in urban fringe areas from the ecological function level. Taking Daxing District, an urban fringe area in Beijing, as an example, the evolution of its landscape pattern from 1993 to 2007 was studied, pointing out that the landscape was severely disturbed by human

activities during urbanization, and the patches tended to be dispersed and fragmented, with increasing separation and fragmentation [18]. He Jianhua [19] studied the impact of land-use change on habitat quality in Ezhou City from two aspects of habitat patch landscape pattern and ecological network connectivity, and called for improving ecological protection of urban fringe areas in metropolitan cities. Zhaolin Wang improved the CA-Markov model to simulate and predict land-use patterns in Beibei District, Chongqing in 2030, with continuing reductions in arable land and water areas, with some implications for sustainable development [20]. With the increasing attention paid to landscape ecology, the sub-issues of landscape ecology and people's quality of life and livability in urban fringe areas have become more and more hotspots in current academic research. Thus, the landscape pattern of the urban fringe promotes the healthy development of the city through ecological self-circulation, and ecology is the most important attribution of the urban fringe.

Wuhan, a city in the middle and lower reaches of the Yangtze River plain, is known as the "thoroughfare of nine provinces" and is a transportation hub in the Chinese hinterland. In 2021, the First Financial Weekly included Wuhan in its list of new first-tier cities, indicating that the city is highly representative of the cities in central China. Since 2000, Wuhan has experienced rapid urbanization, rapid population growth, and continuous redistribution of land resources [21]. The built-up area of the city has been expanding outward, and the expansion pattern is both homogeneous and unique compared to other Chinese cities such as Beijing, Guangzhou and Changchun, and a large amount of agricultural land and ecological land has been occupied [22]. With the expansion of the central city of Wuhan, the ecological environment pressure of the urban fringe is great, and the contradiction between urbanization and the ecological environment is aggravated, which requires further combing and coordination. The fringe area carries traces of urban expansion, but not much attention has been paid to it. In addition, most of the current studies on landscape patterns in the academic community have been carried out for cities or research areas such as park green areas, while there are fewer research results related to the landscape ecological functions of Wuhan's urban fringe area in the past 20 years. This paper analyzes the development process of Wuhan urban fringe areas from 2000 to 2020, delineates the urban fringe areas, and discusses the dynamic change of urban land-use and landscape patterns in order to further reveal the spatial expansion law of Wuhan City and provide data and strategic support for exploring the benign development of Wuhan urban fringe areas.

## 2. Overview of the Research Area

Located between 29 degrees 58 and 31 degrees 22 min north latitude and 113 degrees 41 to 115 min east longitude, Wuhan City has jurisdiction over Jiangan District, Jianghan District, Qiaokou District, Hanyang District, Wuchang District, Qingshan District, Hongshan District, Caidian District, Jiangxia District, Huangpi District, Xinzhou District, East-West Lake District, and 13 administrative districts in Hannan District, covering a total area of 491 km$^2$ (Figure 1). From 2000 to 2021, Wuhan's built-up area expanded from 210 km$^2$ to 885 km$^2$ kilometers, its household registration population increased from 4.41 million to 9.34 million, and its urbanization rate rose to 84.56%. Wuhan is the largest megacity in central China and one of the seven central cities in mainland China. It has experienced rapid urbanization over the past 20 years and has typical characteristics of China's urban development.

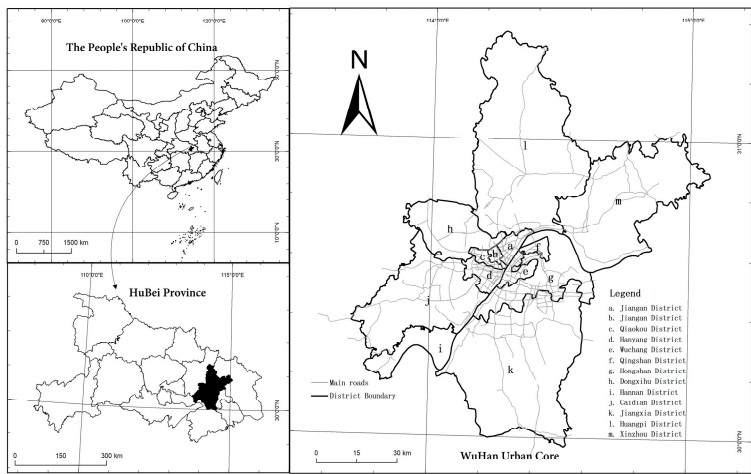

**Figure 1.** Research area.

## 3. Data and Methods

### 3.1. Data Source and Processing

The data used in this paper include:

(1) Data on land-use in Wuhan from 2000 to 2020 (Table 1).

**Table 1.** Data Source.

| Type | Data | Data Source | Format | Year |
|------|------|-------------|--------|------|
| Geographic information data | Land use data | The 30 m annual China land cover dataset (CLCD) https://doi.org/10.5281/zenodo.5210928 (accessed on 7 March 2022) | Raster data | 2000 2010 2020 |
| Population data | Population density | WorldPop 100 m Population density (https://hub.worldpop.org/geodata/listing?id=69) (accessed on 7 March 2022) | Raster data | 2000 2010 2020 |

The land-use/cover data used in this paper were from the China Land Cover Dataset (CLCD) published by Wuhan University based on the Landsat series with a spatial resolution of 30 m from 1990 to 2020. The data is combined with stable samples and satellite time-series data from China land-use/cover dataset (CLUD) and visual interpretation samples from Google Earth and Google Maps for training sample collection, constructs a certain number of time metrics from all available Landsat data, and it is input into a random forest classifier to obtain classification results, and further combines post-processing methods of spatiotemporal filtering and logical reasoning to improve the spatiotemporal consistency of CLCD (Refer to this article for the specific production process) [23]. Classification results include cropland, forest, shrub, grassland, water, snow/ice, barren, impervious surface, and wetland. The years selected for land-use/cover data in this article are 2000, 2010, and 2020.

(2) Wuhan population data from 2000 to 2020.

The Population data for this paper were obtained from WorldPop for the years 2000, 2010, and 2020 at a resolution of 100 m. The population density datasets for all countries of the world for each year 2000–2020 were derived from the corresponding 2000–2020 corresponding set of population statistics for each country for 2000–2020, based on which the data set was adjusted and corrected by the United Nations. The data sets are raster data, with 3 arcseconds (about 100 m at the equator) as the overall raster resolution.

### 3.2. Identification and Analysis Method of Urban Fringe Area

LHRusswurm, a famous geographer, proposed a regional urban model from the perspective of linking urban areas and rural hinterlands in his 1975 "LHRusswurm Model of Marginal Communities" [4], which suggests that regional cities are divided into four parts from the center to the fringe, namely urban core, urban fringe, urban influence, and rural hinterland. Drawing on the LHRusswurm regional city model, the city of Wuhan is divided into the urban core, urban fringe, urban impact area, and rural hinterland from the center to the fringe [15,16]. Since the dynamics of the urban fringe area of Wuhan, the subject of this paper, is highly complex, the urban influence area is included in the urban fringe area for this study.

As a socio-economic multidimensional body, it is one-sided to quantify urban patterns using only a single socio-economic statistic or geospatial data. The indicators can be grouped into the following three dimensions: (1) demographic measures (such as population density, population growth rate, employment rate, ethnic agglomeration, etc.), (2) physical spatial measures (such as road density, construction land share, vegetation, or impervious surface ratio, etc.), (3) landscape measures (such as patch density, fractal dimension, landscape flocculation, etc.). On the one hand, the urbanization process has changed the urban landscape pattern, producing more impervious surfaces and artificial landscapes, resulting in fragmentation of built-up land patches and mixed land use in urban fringe areas. On the other hand, the increase in land-use intensity has brought about an increase in population density in urban fringe areas, but its aggregation is low, making the population density in urban fringe areas higher than that in rural areas but lower than that in urban areas. Summing up the above analysis, and considering the objectivity and applicability of the measurement indexes comprehensively, the impervious surface cover is selected in the physical spatial metric and landscape metric, respectively [24], and landscape flocculation [6,25–27] as the main discriminant, population density in population metric [28] as an auxiliary indicator to define urban fringe areas, as shown in the workflow diagram (Figure 2).

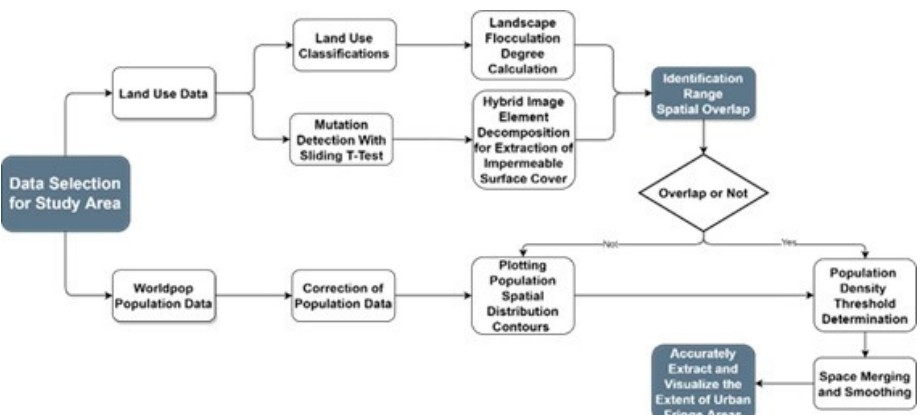

**Figure 2.** Workflow diagram.

### 3.2.1. Impervious Surface Ratio and Moving *t*-Test Technique

Impervious surface is an important indicator of urbanization. From any complete urban–rural spatial cross-section, the closer the area to the urban core, the greater the impervious surface ratio, and the closer the area to the rural hinterland, the smaller the impervious surface ratio, and the impervious surface ratio fluctuates sharply with the distance from the urban center. By extracting the spatially abrupt changes in the impermeability ratio, the urban fringe can be determined preliminarily.

Moving *t*-test is a statistical method that detects mutations by examining significant differences between methods for two random samples. If the differences between methods for two parts of a subsequence exceed a certain significance, the mean can be considered to

have mutated. For a series with $n$ sample sizes, a certain moment is artificially set as the reference point, and the two subseries before and after the reference point $x_1$ and $x_2$ the sample lengths are $n_1$ and $n_2$, the mean values of the two subsequences are $\overline{x_1}$ and $\overline{x_2}$ with variances of $s_1^2$ and $s_2^2$. The expressions are:

$$t = \frac{\overline{x_1} - \overline{x_2}}{S\sqrt{\frac{1}{n_1} + \frac{1}{n_2}}} \tag{1}$$

In

$$S = \sqrt{\frac{n_1 s_1^2 + n_2 s_2^2}{n_1 + n_2 - 2}} \tag{2}$$

The equation obeys the degrees of freedom $\gamma = n_1 + n_2 - 2$ of the distribution. When the points on the $t$-statistic curve exceed the $t_a$ value, then a mutation has occurred.

### 3.2.2. Landscape Flocculation Degree

Information entropy is a physical quantity used to measure the complexity and equilibrium of a system, and it is introduced into the study of land-use landscape patterns to establish the landscape flocculence equation.

$$W = -\sum_{i=1}^{n} XiLN(Xi) \tag{3}$$

where $W$ is the landscape flocculation value, $X_i$ denotes the ratio of a certain type of site to grid cells per unit area, and $I$ denotes the number of site types within a grid cell.

Landscape flocculation reveals the homogeneity and heterogeneity of landscape space and quantifies the degree of fragmentation and fragmentation of the urban landscape. The more land-use types there are in a unit area, the higher the heterogeneity of land-use patches, and the greater the landscape flocculation. The urban core and rural hinterland are mostly large built-up areas or agricultural land, with a relatively homogeneous land-use type and low landscape flocculation. In contrast, the most important feature of the urban fringe is the diversity of land-use types, where various land-use types are interspersed and the layout is loose, and the landscape flocculation degree is high. Therefore, the scope of urban fringe areas can be identified according to the urban–rural spatial differences in landscape flocculation [3].

### 3.2.3. Population Density

Population density is a more direct indicator of the existence of cities as an agglomeration phenomenon. At the same time, population density is a good indicator of the intensity of economic and social activity in cities [3]. The population density of the urban fringe is higher than that of the rural hinterland and lower than that of the urban core. Urban fringe areas have a higher population density than rural hinterlands, lower than urban core areas. Moreover, as distance increases, the population density of urban centers tends to decline rapidly, while the population density gradient in rural hinterlands is very flat [29]. Li Jintao [30] and Chen Wanxu [31] showed that there is a close relationship between population change and urban construction land evolution, and Zhou Xiaochi [6] and Dai Junjie [7] both used population density as one of the criteria for delineating fringe areas. The population density is one of the criteria for the delineation of the edge zone. Based on this, the concept of gradient change of population density can be used to assist in the identification of urban fringe areas.

### 3.3. Analysis of the Method of Land-Use Evolution in the Urban Fringe Area

In this paper, two methods of dynamic land-use change degree and land-use transfer matrix are used to analyze the characteristics of land-use change in urban fringe areas.

3.3.1. Dynamic Change Degree of Land-Use

The calculation model of land-use dynamic change degree represents the dynamic change of various types of land-use within a period, and its expression is:

$$K = \frac{u_b - u_a}{u_a \times t} \times 100\% \tag{4}$$

In the formula, $K$ is the dynamic change degree of a certain land-use type in the study area during the study period $t$, $u_a$ and $u_b$ represent the area of a certain land-use type at the study period, and at the end of the study period, respectively, $t$ is the study period. The positive and negative $K$ values represent the positive and negative changes of a certain land-use type, respectively. When the $K$ value is positive, it means that the area of the land-use type increases during the study period $t$, and a negative $K$ value means a decrease in area.

3.3.2. Land-Use Transition Matrix

The land-use transfer matrix quantitatively describes the mutual conversion area of each land-use type in a certain period, and its content includes the area of a land-use type converted to other land-use types and the conversion of other land-use types to this type of land-use type in this period. The area of the land-use type is divided into two parts. Its expression is:

$$S_{ij} = \begin{bmatrix} S_{11} & S_{12} & \cdots & S_{1n} \\ S_{21} & S_{22} & \cdots & S_{2n} \\ \vdots & \vdots & \vdots & \vdots \\ S_{n1} & S_{n2} & \cdots & S_{nn} \end{bmatrix} \tag{5}$$

In the formula, $S$ is the area, $n$ is the land-use type, $i$ is the land-use type at the beginning of the research period, and $j$ is the land-use type at the end of the research period.

*3.4. Landscape Pattern Index Analysis Method*

The Landscape index is a commonly used spatial analysis method to quantitatively express the relationship between landscape patterns and ecological processes. It highly condenses landscape pattern information and reflects some aspects of its composition and spatial configuration. It is a relatively mature quantitative research method in landscape pattern research [7,15,18].

In this paper, Fragstats is used as a technology platform to calculate and analyze landscape patterns from two levels: type landscape index and horizontal landscape index. Combining the existing research results and the actual situation, the selected landscape type indices include: perimeter area fractional dimension (PAFRAC), maximum patch index (LPI), average patch size (AREA_MN), patch type area (CA), patch density (PD), and the number of patches (NP); the selected landscape horizontal level indices include: shape index (LSI), Shannon evenness Index (SHEI), Shannon diversity index (SHDI), aggregation index (AI), and patch density (PD) as shown in Table 2. The land-use data of urban fringe areas in Wuhan for three periods of 2000, 2010, and 2020 were converted to Grid format and imported into Fragstats to select the above landscape pattern indices for calculation.

**Table 2.** Landscape pattern index select.

| Direction | Indicator Name | Index Name |
|---|---|---|
| Type-level landscape pattern index | Indicator name | Perimeter-area fractal dimension (PAFRAC) |
| | Shape metric | Largest patch index (LPI) |
| | Area and edge metric | Mean of patch area (A REA_MN) |
| | | Class area (CA) |

**Table 2.** *Cont.*

| Direction | Indicator Name | Index Name |
|---|---|---|
| Horizontal-level landscape pattern index. | Aggregation metric | Number of patches (NP) |
| | | Patch density (PD) |
| | | Landscape shape index (LSI) |
| | Shape metric | Shannon evenness index (SHEI) |
| | | Shannon diversity index (SHDI) |
| | | Aggregation index (AI) |
| | | Patch density (PD) |

## 4. Results and Analysis

### 4.1. Identification of Urban Fringe Areas

In this paper, the impermeable surface ratio and landscape flocculation of Wuhan City are determined, and the population density threshold is adopted to eliminate the accidental factors in the scope delimitation to the maximum extent and improve the scientific character of the urban fringe area delimitation.

#### 4.1.1. Extraction of Impervious Surface Ratio Indicators

To extract the extent of urban fringe areas through impervious surface ratio, the key is to extract 2 mutation points of impervious surface ratio, i.e., the inner ring mutation point from the urban core to the urban fringe area and the outer ring mutation point from the urban fringe area to rural hinterland [25,32]. In this paper, the mutation points of impermeable surface ratio are extracted by moving *t*-test. Its operation steps include the design of spatial sampling points, the acquisition of mutation points, the extraction of mutation points, and the description of the boundary of the urban fringe areas.

(1) Design of spatial sampling points.

First, take the Jianghan Road shopping area as the city center, and draw 360 section lines at an angle of 1° in ArcGIS. Then, form two kinds of concentric circle with a radius of 500 m and 1000 m, covering the entire study area. Concentric circles and section lines are intersected to get spatial sampling points, and the two are intersected to get spatial sampling points. The process of extracting masks and comparing data with impermeable ground ratios to obtain data points covering the whole Wuhan City area is shown in Figure 3.

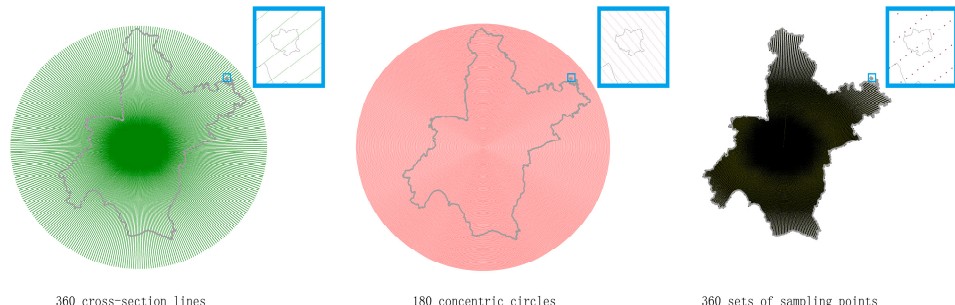

360 cross-section lines      180 concentric circles      360 sets of sampling points

**Figure 3.** Spatial sampling point design process.

(2) Acquisition of mutation sites.

The data points were divided into 360 groups of data columns in MATLAB, and a moving *t*-test was performed on each group of data columns in step 5 to obtain mutation points. Figure 4 shows the test results for some of the data series, and the data points corresponding to the range beyond the yellow dashed line and the purple dashed line in the figure are the mutation points. After comparison, the mutation points obtained at a radius difference of 500 m are more accurate than those obtained at a radius difference of

1000 m. Therefore, the next step is performed based on the mutation points obtained from concentric circles with a radius difference of 500 m.

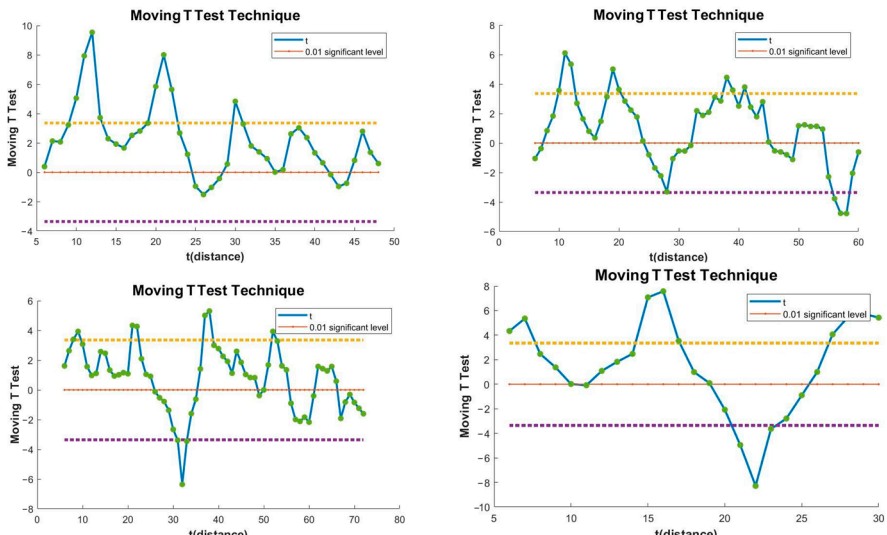

**Figure 4.** Partial data sequence test results.

(3) Extraction of mutation points and depiction of urban fringe boundaries.

To extract and visualize the mutation points on 360 data columns, download 2000, 2010, and 2020 satellite image maps of Wuhan City in Google Earth, manually remove the abnormal mutation points according to the satellite image maps, and finally identify the mutation points on the boundary between urban core area and urban fringe area and the mutation points on the boundary between the urban fringe area and rural hinterland in 360 directions, and connect them in order. The inner and outer boundaries of the urban fringe area are obtained by sketching, and the spatial continuity and integrity of the area should be maintained in the depiction process, and finally, the results are appropriately corrected, and the results are shown in Figure 5.

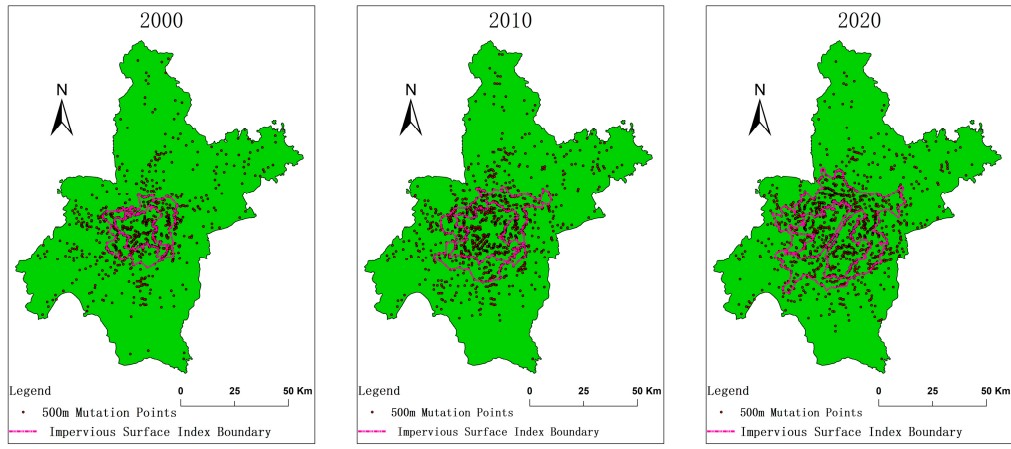

**Figure 5.** Impervious surface ratio of Wuhan City 2000–2020 extracted urban fringe area range.

After the above process, the range of urban fringe areas in Wuhan from 2000 to 2020 can be obtained (Figure 5). Since urban fringe areas are extremely dynamic, fuzzy, and uncertain, it is not universally meaningful to delineate the range of urban fringe areas from a single aspect, so this paper will use the extraction of landscape flocculence to further delineate the range of urban fringe areas in Wuhan in order to improve its accuracy.

### 4.1.2. Extraction of Landscape Flocculation Index

The land-use cover data of Hubei Province for three periods of 2000, 2010, and 2020 were downloaded from Zenodo and cropped using the administrative boundary vector data of Wuhan City to obtain the land cover data of Wuhan City for three periods. There are seven types of land use: agricultural land, forest land, impermeable surface, unused land, water body, shrub, and grassland. Since too many land classifications reduce the accuracy of urban fringe identification, land cover types are reclassified in determining the boundaries of urban fringe areas, and land close to rural hinterland characteristics, such as agricultural land, forest land, and grassland, are merged into agricultural land, while other land-use types remain unchanged. A 500 m × 500 m grid area was established, and each land type was allocated to each net through an area tabulation tool to calculate the proportion of each land type in each net. As can be seen from Figure 6, the spatial variability of landscape flotsam is significant, showing differences in landscape heterogeneity between urban centers, rural hinterlands, and urban margins. In the impervious surface and large agricultural land areas, the landscape flocculation is low, varying between 0 and 0.7. Areas in the urban fringe and on the edges of water bodies have higher values than the urban and rural areas. After several trials, it was determined that a large number of areas surrounding the main urban area with information entropy values of 0.7 and above would be the extent of the urban fringe.

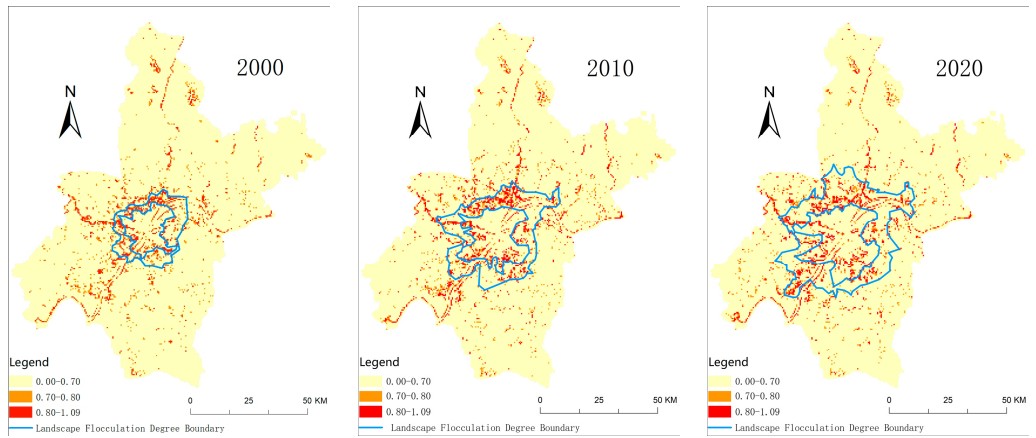

**Figure 6.** The spatial division of landscape flocculation in the urban fringe of Wuhan, 2000–2020.

### 4.1.3. Determination of Population Density Indicators

First is the demographic correction. WorldPop raster represents the total number of populations within the range expressed by each grid. The population numbers for 2000, 2010, and 2020 were calculated to be 863.76, 960.03, and 11,491.2 thousand people respectively using the tabular display partition statistics tool, and the resident population data for the fifth, sixth, and seventh censuses in Wuhan were 831.20, 978.53, and 12,326.5 thousand people, respectively, and were multiplied by correction factors of 0.9623, 1.0193, and 1.0726 to get the corrected numbers. Since the data is at $100 \times 100$ m$^2$ resolution, which is more representative of the population distribution in the landscape than the administrative units, the spatialization of population density is not performed here. Secondly, the data at $100 \times 100$ m$^2$ resolution maximizes the impact of the inflated population density gradient due to the large proportion of undeveloped land in the outer urban areas on the study results. Finally, the population density contours were drawn to determine the population threshold range in the urban fringe, which was used to correct the previously delineated urban fringe range. After several iterations of corrections and adjustments, the population density greater than 3000 persons/km$^2$ is close to the inner edge of the city, and the population density of 1000–3000 persons/km$^2$ is closer to the outer urban edge.

### 4.1.4. Determining the Extent of the Urban Fringe

The urban fringe areas of Wuhan City extracted from impervious surface mutation points and landscape flocculation degrees were subjected to spatial overlay operation. Compare the urban fringe determined by the impervious surface ratio and landscape flocculation, for the overlapping areas, they can be directly divided into Wuhan urban fringe areas; for the non-overlapping areas, i.e., those with only one indicator judged as urban fringe areas, the population density threshold is used for screening, and the scope of Wuhan urban fringe areas in the three periods of 2000, 2010 and 2020 is finally determined (Figure 7). During the third period, urban fringe areas of Wuhan were 474.351 km$^2$, 864.982 km$^2$, and 1220.104 km$^2$, respectively. With the increase in the urban core area, the urban fringe area gradually expanded. As can be seen from the graph, the circle structure of Wuhan in the urban core, urban fringe, and rural hinterland has become very obvious in the last three years.

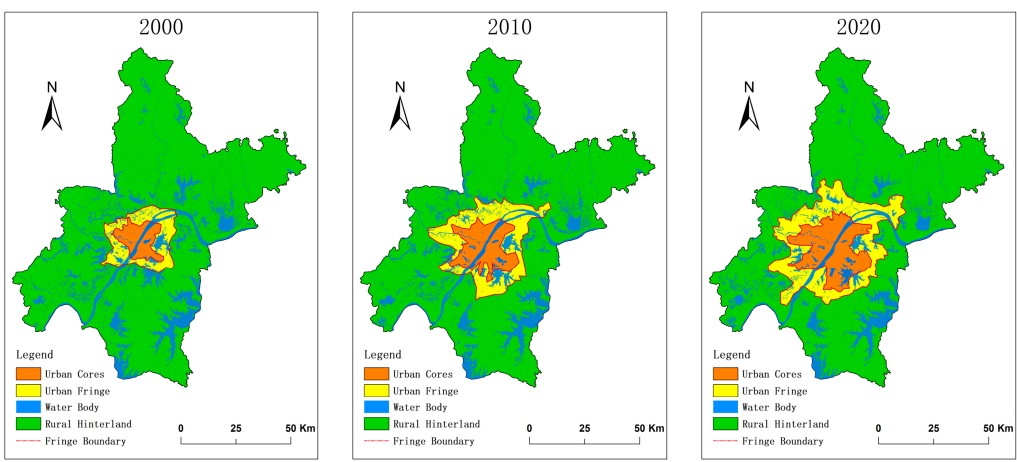

**Figure 7.** Spatial distribution of urban fringe areas in Wuhan from 2000 to 2020.

### 4.2. *Analysis of the Spatial Characteristics of Urban Fringe Expansion*

It is clear from the Figure 7 that Wuhan City shows a clear circle structure of the urban core, urban fringe, and rural hinterland in all three years.

According to Tables 3 and 4, from 2000 to 2010, the area of Wuhan's urban core zone grew by 230.27 km$^2$, an increase of 90.81%, with an expansion rate of 23.03 km$^2$/a. The area of Wuhan's urban fringe zone grew by 390.63 km$^2$, an increase of 82.35%, with an average annual expansion of 39.06 km$^2$. From 2010 to 2020, the area of the core zone grew by 260.74 km$^2$, with an expansion rate of 53.89% and an average annual expansion of 26.07 km$^2$. The area of the urban fringe zone grew by 355.12 km$^2$, with an increase of 41.06% and an expansion rate of 35.53 km$^2$/a. ① The urban core area and the urban fringe area of Wuhan expand rapidly from 2000 to 2020, and the expansion area of the urban fringe area is 254.75 km$^2$ larger than that of the urban core area; ② the area of Wuhan City expanded more and the expansion rate was faster from 2000 to 2010, and the development of Wuhan City was extremely rapid from 2000 to 2010; ③ the urban core area and the urban fringe area still maintained the expansion trend from 2010 to 2020, but the expansion ratio of both decreased significantly, and the expansion ratio of the urban core area in the latter stage was 0.6 times of the previous stage, and 0.5 times of the previous stage in the latter stage of the urban fringe area; in addition, the expansion rate and expansion area of the urban fringe area also gradually decreased.

**Table 3.** Area statistics for the geographic spatial structure of Wuhan, 2000–2020.

| Statistical Table of the Area of the Geospatial Structure | | | |
|---|---|---|---|
| Sports event | Area/km$^2$ by year | | |
| | 2000 | 2010 | 2020 |
| City center district | 253.573 | 483.840 | 744.575 |
| Urban fringe areas | 474.351 | 864.982 | 1220.104 |
| Rural hinterland | 7853.823 | 7232.925 | 6617.068 |

**Table 4.** Number of urban fringe expansions in Wuhan, 2000–2020.

| | Urban Core | | | Urban Fringe | | |
|---|---|---|---|---|---|---|
| | Extended Area (km$^2$) | Expansion Ratio (%) | Expansion Rate (km$^2$/a) | Extended Area (km$^2$) | Expansion Ratio (%) | Expansion Rate (km$^2$/a) |
| 2000–2010 | 230.27 | 90.81 | 23.03 | 390.63 | 82.35 | 39.06 |
| 2010–2020 | 260.74 | 53.89 | 26.07 | 355.12 | 41.06 | 35.51 |
| 2000–2020 | 491.00 | 193.63 | 24.55 | 745.75 | 157.22 | 37.29 |

### 4.2.1. Extended Types

The expansion of urban fringe areas is divided into three types: outward expansion type, internal infill type, and conversion core type, which are manifested as the transformation of rural hinterland into urban fringe area, the transformation of regional scope unchanged but internal land-use and other changes, and the transformation of urban fringe area into urban core area [24].

In the process of urban development in Wuhan from 2000 to 2010, the area transformed from rural hinterland to the urban fringe area in Wuhan was 583.24 km$^2$, accounting for 67.43% of the urban fringe area in 2010, in the stage of outward expansion; 279.81 km$^2$ of the regional territorial spatial structure did not change and remained as urban fringe area, accounting for 32.35%, in the stage of internal filling; the area of urban fringe area transformed into the urban core area is 194.54 km$^2$, in the stage of transforming core. It is known that from 2000 to 2010, the types of expansion in the urban fringe of Wuhan were mainly outward expansion types (Table 5).

**Table 5.** Number of geospatial structure transfers in Wuhan, 2000–2010.

| Geographical Spatial Structure shift in Wuhan from 2000 to 2010 (Unit: km$^2$) | | | | |
|---|---|---|---|---|
| | 2010 Geographical Spatial Structure | | | |
| Geographical Spatial Structure in 2000 | Urban Core | Urban Fringe | Rural Hinterland | Aggregate |
| Urban core | 251.63 | 1.94 | 0.00 | 253.57 |
| Urban fringe | 194.54 | 279.81 | 0.00 | 474.35 |
| Rural hinterland | 37.66 | 583.24 | 7232.92 | 7853.82 |
| Aggregate | 483.84 | 864.98 | 7232.92 | 8581.75 |

From 2010 to 2020, the area in the outward expansion phase is 615.46 km$^2$, accounting for 50.44% of the urban fringe area in 2010. The area in the internal infill phase is 599.75 km$^2$, accounting for 49.16% of the urban fringe area in 2010. The area in the transformation core phase is 263.06 km$^2$ (2010–2020 Wuhan City). The main expansion types of urban fringe areas are outward expansion type and internal infill type (Table 6).

**Table 6.** Changes in land-use types in Wuhan from 2000 to 2020.

| Geographical Spatial Structure Shift in Wuhan from 2000 to 2010 (Unit: km²) | | | | |
|---|---|---|---|---|
| **Geographical Spatial Structure in 2010** | **2020 Geographical Spatial Structure** | | | **Aggregate** |
| **Types** | **Urban Core** | **Urban Fringe** | **Rural Hinterland** | |
| Urban core | 478.84 | 5.00 | 0.00 | 483.84 |
| Urban fringe | 263.06 | 599.75 | 2.18 | 864.98 |
| Rural hinterland | 2.68 | 615.36 | 6614.89 | 7232.92 |
| Aggregate | 744.57 | 1220.10 | 6617.07 | 8581.75 |

From the above, it can be seen that the urban fringe area of Wuhan developed rapidly from 2000 to 2010, mainly manifesting itself in outward expansion, while the expansion of the urban core area was slow. The difference between the proportion of internal infill area and the proportion of outward expansion area from 2010 to 2020 is not large, indicating that the urban fringe area maintained its expansion trend during this period while its internal development was also gradual, while the expansion of the urban core area remained slow.

4.2.2. Extended Directions

The main expansion directions of the urban fringe area in Wuhan from 2000 to 2020 are five directions: southwest, west, north, northeast, and east (Figure 8), and the increment of the urban fringe area accounts for 21%, 15%, 15%, 13%, and 13% of the total increment of the fringe area (Table 7), and these five directions can explain 76.2% of the growth of the urban fringe area. From 2000 to 2010, the main growth directions were south, west, and northeast, with 31%, 21%, and 18%, respectively, and southwest, north, and east, with 47%, 20%, and 18%, respectively, from 2010 to 2020.

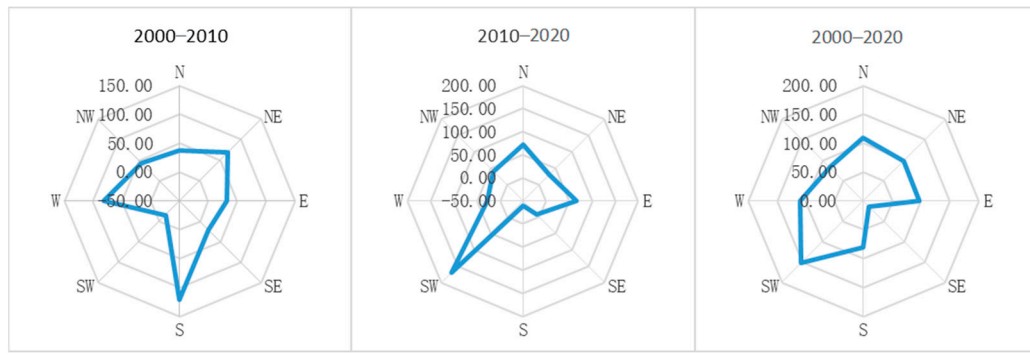

**Figure 8.** Area distribution of Wuhan urban fringe areas along different directions of expansion, 2000–2020.

**Table 7.** Area of urban fringe expansion by direction, Wuhan, 2000–2020.

| | The Urban Fringe Area in Each Direction | | | Increment of the Urban Fringe Area in Each Direction | | |
|---|---|---|---|---|---|---|
| Direction | 2000 | 2010 | 2020 | 2000–2010 | 2010–2020 | 2000–2020 |
| North | 43.52 | 80.65 | 152.50 | 37.13 | 71.85 | 108.98 |
| Northeast | 87.56 | 156.20 | 186.06 | 68.63 | 29.86 | 98.49 |
| East | 83.23 | 115.44 | 180.73 | 32.21 | 65.29 | 97.50 |
| Southeast | 82.33 | 104.33 | 95.80 | 22.00 | −8.53 | 13.48 |
| South | 23.45 | 145.06 | 104.71 | 121.61 | −40.36 | 81.26 |
| Southwest | 81.43 | 66.24 | 234.50 | −15.19 | 168.27 | 153.07 |
| West | 46.06 | 128.18 | 156.13 | 82.13 | 27.95 | 110.08 |
| Northwest | 26.77 | 68.88 | 109.67 | 42.11 | 40.79 | 82.90 |
| Sum | 474.35 | 864.98 | 1220.10 | 390.63 | 355.12 | 745.75 |

Combined with the topographic map of Wuhan, it can also be found that two major rivers, the Hanshui and the Yangtze, traverse the urban area of Wuhan. Moreover, 2000–2010 expansion of the fringe area of Wuhan to the west is consistent with the alignment of the Hanshui. Further, 2000–2010 expansion to the northeast and 2010–2020 expansion to the southwest are also consistent with the alignment of the Yangtze. At the same time, the expansion of the site in all directions is consistent with the direction of the radial external traffic roads. It indicates that topography, water bodies, and traffic arteries, influence the direction of urban expansion.

### 4.2.3. Extended Areas

From 2000 to 2020, the main expansion areas of the urban fringe of Wuhan were Hongshan District, Caidan District, East-West Lake District, Hannan District, Huangpi District, Jiangxia District, and Xinzhou District, as shown in Table 8. The expansion areas were 53.58 km$^2$, 190.45 km$^2$, 114.97 km$^2$, 35.25 km$^2$, 210.81 km$^2$, 150.65 km$^2$, and 86.84 km$^2$, respectively.

**Table 8.** Area of geographical spatial structure within administrative districts of Wuhan City, 2000–2020 (km$^2$).

| Statistical Table of the Geographical Spatial Structure of the Administrative Regions of Wuhan City in Terms of Area (km$^2$) | | | | | | | | | |
|---|---|---|---|---|---|---|---|---|---|
| Administrative district | Urban core | | | Urban fringe | | | Rural hinterland | | |
| | 2000 | 2010 | 2020 | 2000 | 2010 | 2020 | 2000 | 2010 | 2020 |
| Ganglian District of central Chongqing municipality, formerly in Sichuan | 33.27 | 37.57 | 64.96 | 47.79 | 43.48 | 16.09 | 0.00 | 0.00 | 0.00 |
| Jianghan District of central Chongqing municipality, formerly in Sichuan | 27.91 | 28.54 | 28.54 | 0.63 | 0.00 | 0.00 | 0.00 | 0.00 | 0.00 |
| Qiaokou District of Qiaokou, Sichuan | 28.84 | 32.13 | 38.17 | 11.51 | 8.22 | 2.17 | 0.00 | 0.00 | 0.00 |
| Hanyang District of central Chongqing municipality, formerly in Sichuan | 33.47 | 83.45 | 106.00 | 64.20 | 27.83 | 6.59 | 14.92 | 1.31 | 0.00 |
| Wuchang District of Beijing | 68.08 | 69.06 | 73.08 | 27.48 | 31.17 | 27.22 | 4.74 | 0.07 | 0.00 |
| Green Hills District of Beijing | 12.53 | 17.73 | 26.01 | 44.69 | 51.34 | 47.41 | 34.46 | 22.61 | 18.27 |
| Hongshan Suburban District of Hong Kong | 45.48 | 130.37 | 169.86 | 137.61 | 149.98 | 191.19 | 331.49 | 234.24 | 153.54 |
| Caidian District, Hainan | 0.00 | 38.62 | 88.33 | 38.13 | 107.05 | 228.58 | 1051.69 | 944.15 | 772.92 |
| East-West Lake District | 3.98 | 14.72 | 27.32 | 42.89 | 113.41 | 157.86 | 444.77 | 363.52 | 306.46 |
| Han'nan District of central Chongqing municipality, formerly in Sichuan | 0.00 | 0.00 | 0.00 | 0.00 | 0.00 | 35.25 | 289.19 | 289.19 | 253.94 |
| Huangpi District | 0.00 | 0.00 | 0.28 | 26.31 | 112.11 | 237.13 | 2219.10 | 2133.31 | 2008.02 |
| Jiangxia District | 0.00 | 31.65 | 122.04 | 33.12 | 194.46 | 183.77 | 1977.97 | 1784.98 | 1705.28 |
| Xinzhou District | 0.00 | 0.00 | 0.00 | 0.00 | 25.94 | 86.84 | 1477.10 | 1451.16 | 1390.26 |
| Aggregate | 3.98 | 84.99 | 237.96 | 140.46 | 552.97 | 929.42 | 7459.82 | 6966.31 | 6436.88 |

The main expansion areas of the urban core are Jiangan District, Qiaokou District, Jianghan District, Wuchang District, Qingshan District, Hongshan District, Caidian District, East-West Lake District, and Jiangxia District, with 31.7 km$^2$, 9.33 km$^2$, 0.63 km$^2$, 5.0 km$^2$, 13.47 km$^2$, 124.38 km$^2$, 88.33 km$^2$, 23.34 km$^2$, and 122.04 km$^2$ (Figure 9), respectively. Among them, Hanyang District, Hongshan District, Caidian District, and Jiangxia District grew more drastically.

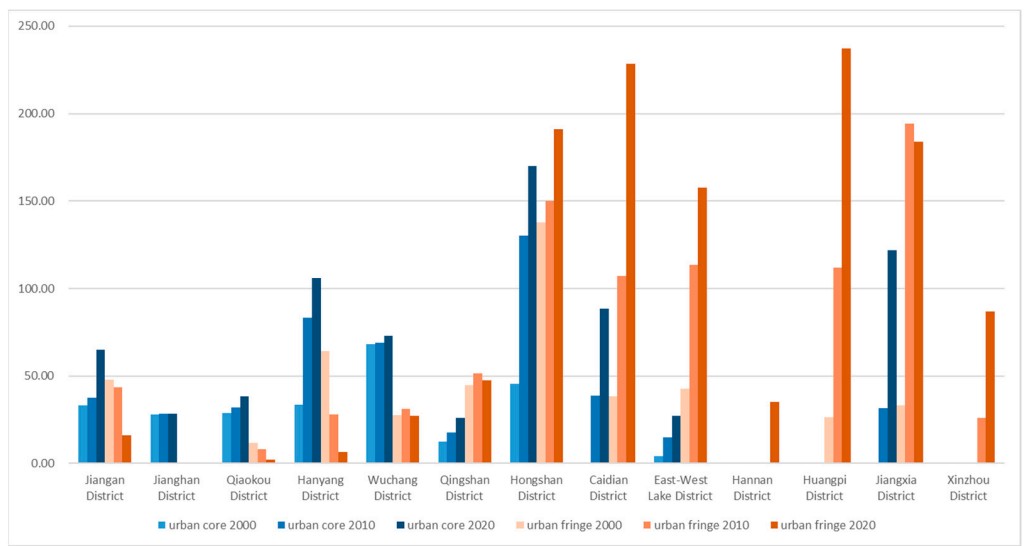

**Figure 9.** Area histogram of the geographical spatial structure within each administrative region of Wuhan, 2000–2020.

From 2000 to 2020, Jianghan District, Jianghan, Qiaokou, Hanyang, Wuchang, Qingshan District, and Hongshan District accounted for most of the region. Analyzing Table 9, we can see that the urban core area of Caidian District and Jiangxia District accounted for 0 in 2000, the urban core area of Caidian District accounted for 12%, and Jiangxia District accounted for 16% in 2020, exceeding some core areas such as Qingshan District, Wuchang District, Jianghan District, Qiaokou District, and Jiangan District, indicating that Wuhan City gradually develops from single core to polycentric gradually. According to the data of the proportion of urban fringe areas in each administrative district from 2000 to 2020, the proportion of urban fringe areas in the core areas of Wuhan has been decreasing year by year, among which Jianghan District and Qiaokou District have been fully developed into urban core areas; only Jiangxia District shows the trend of increasing and then decreasing the proportion of non-core areas, while all other districts show the trend of increasing the proportion. In summary, with urbanization, the core development areas of Wuhan have gradually completed the transformation of urban fringe areas into urban core areas, while most of the non-core areas of the city have expanded into the countryside, transforming the rural hinterland into urban fringe areas, and a few have further transformed urban fringe areas into urban core areas on this basis.

**Table 9.** Share of geographical spatial structure by administrative districts in Wuhan, 2000–2020.

| Percentage of Geographical Spatial Structure by Administrative Districts in Wuhan | | | | | | |
|---|---|---|---|---|---|---|
| Administrative district | urban core | | | Urban fringe areas | | |
| | 2000 | 2010 | 2020 | 2000 | 2010 | 2020 |
| Ganglian District of central Chongqing municipality, formerly in Sichuan | 0.13 | 0.08 | 0.09 | 0.10 | 0.05 | 0.01 |
| Jianghan District of central Chongqing municipality, formerly in Sichuan | 0.11 | 0.06 | 0.04 | 0.00 | 0.00 | 0.00 |
| Qiaokou District of Qiaokou, Sichuan | 0.11 | 0.07 | 0.05 | 0.02 | 0.01 | 0.00 |
| Hanyang District of central Chongqing municipality, formerly in Sichuan | 0.13 | 0.17 | 0.14 | 0.14 | 0.03 | 0.01 |
| Wuchang District of Beijing | 0.27 | 0.14 | 0.10 | 0.06 | 0.04 | 0.02 |
| Green Hills District of Beijing | 0.05 | 0.04 | 0.03 | 0.09 | 0.06 | 0.04 |
| Hongshan Suburban District of Hong Kong | 0.18 | 0.27 | 0.23 | 0.29 | 0.17 | 0.16 |
| Caidian District, Hainan | 0.00 | 0.08 | 0.12 | 0.08 | 0.12 | 0.19 |

**Table 9.** *Cont.*

| Percentage of Geographical Spatial Structure by Administrative Districts in Wuhan | | | | | | |
|---|---|---|---|---|---|---|
| East-West Lake District | 0.02 | 0.03 | 0.04 | 0.09 | 0.13 | 0.13 |
| Hannan District of central Chongqing municipality, formerly in Sichuan | 0.00 | 0.00 | 0.00 | 0.00 | 0.00 | 0.03 |
| Huangpi District | 0.00 | 0.00 | 0.00 | 0.06 | 0.13 | 0.19 |
| Jiangxia District | 0.00 | 0.07 | 0.16 | 0.07 | 0.22 | 0.15 |
| Xinzhou District | 0.00 | 0.00 | 0.00 | 0.00 | 0.03 | 0.07 |
| Aggregate | 1.00 | 1.00 | 1.00 | 1.00 | 1.00 | 1.00 |

*4.3. Spatial and Temporal Dynamics of Land Use in Urban Fringe Areas*

The dynamic attitude analysis and land-use transfer matrix analysis were conducted within the urban geospatial structure to quantify the geospatial structure and spatio-temporal dynamics of land use in Wuhan from 2000 to 2020.

4.3.1. Analysis of Land Use Dynamics

From 2000 to 2020, the urban fringe area of Wuhan City has been expanded, and the calculation shows the various types of land use in the urban core area, urban fringe area, and rural hinterland (Table 10). Over the past 20 years, the area of impermeable land and woodland in Wuhan has been increasing. Of these, the most impermeable area increased by 1177.765 km$^2$, followed by woodland, which increased by 646.856 km$^2$. Grassland, farmland, shrubs, wasteland, and water bodies all show a decreasing trend, with farmland decreasing the most area decrease 5561.842 km$^2$, followed by waterbodies. During this period, Wuhan's economy has been growing and the area of urban land has increased dramatically, of which 90.1% comes from agricultural land and 9% from water bodies.

**Table 10.** Land-use area in Wuhan, 2000–2020 (unit: %).

| Geospatial Structure | | Urban Core | | | Urban Fringe Areas | | | Rural Hinterland | | |
|---|---|---|---|---|---|---|---|---|---|---|
| | Year | 2000 | 2010 | 2020 | 2000 | 2010 | 2020 | 2000 | 2010 | 2020 |
| | impermeable surface | 66.73 | 65.69 | 64.05 | 27.21 | 29.49 | 32.48 | 2.90 | 3.96 | 4.62 |
| | Grassland | 0.00 | 0.06 | 0.01 | 0.00 | 0.11 | 0.06 | 0.04 | 0.04 | 0.01 |
| | Cultivated land | 14.94 | 18.34 | 19.46 | 46.24 | 46.76 | 48.40 | 76.25 | 75.25 | 73.04 |
| Land use | Shrub | 0.00 | 0.00 | 0.00 | 0.00 | 0.00 | 0.00 | 0.00 | 0.00 | 0.00 |
| | Wasteland | 0.00 | 0.01 | 0.01 | 0.00 | 0.01 | 0.02 | 0.01 | 0.00 | 0.00 |
| | Woodland | 0.39 | 0.35 | 0.84 | 1.40 | 2.18 | 2.77 | 6.40 | 6.87 | 9.18 |
| | Water | 17.95 | 15.59 | 15.62 | 25.15 | 21.45 | 16.27 | 14.40 | 13.88 | 13.16 |
| | Aggregate | 100.0 | 100.0 | 100.0 | 100.0 | 100.0 | 100.0 | 100.0 | 100.0 | 100.0 |

This paper uses the dynamic attitude model to analyze the change of land types in Wuhan City, quantifying the magnitude and speed of change of each land type, and the results are shown in Table 11. from the dynamic attitude of land use, the impervious surface dynamic change is the most intense in 2000–2020, with a dynamic attitude of 6.210%, and the total area increased by 676.549 km$^2$, 2000–2010 The single dynamic attitude of impervious surface was 6.343% in 2010, during this period Wuhan City developed rapidly and the area of impervious surface increased greatly, the dynamic attitude of impervious surface was 3.718% in 2010–2020, indicating that in this period Wuhan City slowed down its development and gradually carried out a stable development stage; the shrub dynamic change degree was the second, 4.355%, the shrub area changed less, but due to its small base, it makes the dynamic attitude larger; the dynamic attitude of woodland is 1.346%, with an area increase of 137.202 km$^2$ and an average annual increase of 6.86 km$^2$, thanks to Wuhan's long-term ecological restoration policy. The dynamic attitude of grassland is

3.013%, with an area decrease of 1.889 km$^2$ and an average annual decrease of 0.094 km$^2$; the dynamic attitude of farmland is 0.542%, with an area decrease of 676.548 km$^2$ and an average annual decrease of 33.827 km$^2$. The motility of wasteland was 3.230% with an area decrease of 0.677 km$^2$ and an average annual decrease of 0.034 km$^2$; the motility of water bodies was 0.427% with a decrease of 110.5 km$^2$ and an average annual decrease of 5.525 km$^2$. In general, the area of agricultural land and water bodies in the urban fringe of Wuhan City has been reduced substantially in the last two decades, and the impervious area has increased substantially.

**Table 11.** Changes in land-use types in Wuhan, 2000–2020.

| Year | Variations | Impermeable Surface | Grassland | Cultivated Land | Shrub | Wasteland | Woodland | Water Column |
|---|---|---|---|---|---|---|---|---|
| 2000–2010 | Magnitude of change in area/km$^2$ | 333.207 | 0.841 | −309.481 | −0.012 | −0.787 | 7.350 | −31.119 |
| | Average annual change | 33.321 | 0.084 | −30.948 | −0.001 | −0.079 | 0.735 | −3.112 |
| | Single-motion attitude/% | 6.343 | 2.684 | −0.496 | −4.196 | −7.517 | 0.144 | −0.240 |
| 2010–2020 | Magnitude of change in area/km$^2$ | 319.229 | −2.730 | −367.068 | −0.013 | 0.111 | 129.851 | −79.381 |
| | Average annual variation | 31.923 | −0.273 | −36.707 | −0.001 | 0.011 | 12.985 | −7.938 |
| | Single-motion attitude/% | 3.718 | −6.867 | −0.619 | −7.776 | 4.256 | 2.512 | −0.628 |
| 2000–2020 | Magnitude of change in area/km$^2$ | 652.437 | −1.889 | −676.549 | −0.024 | −0.677 | 137.202 | −110.500 |
| | Average annual change | 32.622 | −0.094 | −33.827 | −0.001 | −0.034 | 6.860 | −5.525 |
| | Single-motion attitude/% | 6.210 | −3.013 | −0.542 | −4.355 | −3.230 | 1.346 | −0.427 |

### 4.3.2. Analysis of Land Use Shifts

In order to specifically analyze the conversion of various land types in Wuhan during the time span of 2000–2020, this paper analyzes the land-use classification of Wuhan from 2000–2010 and 2010–2020 superimposed on each other to obtain the land-use transfer matrix for the two periods. The Spatiotemporal evolution process of the urban fringe area superimposed on the land-use transfer classification is extremely complex, so the transformation of each type of land in the urban core, urban fringe, and rural hinterland is discussed in the body of this paper, and the detailed table is shown in the Appendix A Tables A1–A4.

From Table 12, it can be seen that during 2000–2010, in terms of land outflow, 194.371 km$^2$ of the urban fringe area was transferred to the urban core, and in terms of land outflow, it was mainly agricultural land, impervious surface and water bodies, with a conversion area of 99.976 km$^2$, 62.309 km$^2$, and 42.024 km$^2$, respectively, followed by forest land, grassland, and wasteland. The main outflow object of the rural hinterland is the urban fringe area, with an outflow area of 582.724 km$^2$, including 377.647 km$^2$ from agricultural land, 142.234 km$^2$ from water bodies, 48.721 from the impervious surface, and a small amount of forest land and grassland. The area of the rural hinterland flowing into the urban core is smaller, 37.627 km$^2$, mainly from agricultural land and water bodies. Matrix (unit: km$^2$)

During 2010–2020, 262.820 km$^2$ of the urban fringe area was transferred into the urban core, with the main types of land transferred being agricultural land, impervious surface and water bodies, with 121.867 km$^2$, 86.343 km$^2$, and 50.544 km$^2$ transferred, followed by forest land (3.843 km$^2$), grassland (0.214 km$^2$), and wasteland (0.01 km$^2$), while only 2.718 km$^2$ of the urban fringe area was transferred to the rural hinterland (Table 13). The rural hinterland flows into the urban core with a smaller area of 2.675 km$^2$, mainly from agricultural land, woodland, and water bodies, and into the urban fringe with 614.779 km$^2$, mainly agricultural land, water bodies, and impervious surface with 422.946 km$^2$, 94.949 km$^2$, and 80.742 km$^2$, respectively, followed by woodland (15.582 km$^2$), grassland (0.544 km$^2$), and wasteland (0.014 km$^2$).

**Table 12.** Urban geographical structure land-use transfer in Wuhan City, 2000–2010.

| Geographical Spatial Structure in 2000 | Land Type | Geographical Spatial Structure in 2010 | | | Aggregate |
|---|---|---|---|---|---|
| | | Urban Core | Urban Fringe | Rural Hinterland | |
| Urban core | Impermeable surface | 168.506 | 0.544 | 0 | 169.051 |
| | Grassland | 0.008 | 0 | 0 | 0.008 |
| | Cultivated land | 36.895 | 0.948 | 0 | 37.843 |
| | Wasteland | 0.005 | 0 | 0 | 0.005 |
| | Woodland | 0.976 | 0.003 | 0 | 0.979 |
| | Water column | 45.025 | 0.443 | 0 | 45.468 |
| | Aggregate | 251.416 | 1.938 | 0 | 253.354 |
| Urban fringe | Impermeable surface | 62.309 | 66.659 | 0 | 128.969 |
| | Grassland | 0.007 | 0.013 | 0 | 0.021 |
| | Cultivated land | 88.876 | 130.298 | 0 | 219.173 |
| | Wasteland | 0.001 | 0 | 0 | 0.001 |
| | Woodland | 1.154 | 5.459 | 0 | 6.613 |
| | Water column | 42.024 | 77.153 | 0 | 119.178 |
| | Aggregate | 194.371 | 279.583 | 0 | 473.954 |
| Rural hinterland | Impermeable surface | 5.302 | 48.721 | 173.286 | 227.309 |
| | Grassland | 0 | 0.170 | 2.935 | 3.105 |
| | Cultivated land | 21.914 | 377.647 | 5581.813 | 5981.375 |
| | Shrub | 0 | 0 | 0.028 | 0.028 |
| | Wasteland | 0 | 0 | 1.041 | 1.041 |
| | Woodland | 0.039 | 13.951 | 488.072 | 502.062 |
| | Water column | 10.372 | 142.234 | 977.235 | 1129.842 |
| | Aggregate | 37.627 | 582.724 | 7224.411 | 7844.762 |
| Aggregate | | 483.415 | 864.245 | 7224.411 | 8572.071 |

**Table 13.** Land-use transfer matrix for the urban geographic structure of Wuhan City, 2010–2020 (unit: km$^2$).

| Geographical Spatial Structure in 2010 | Land Type | Geographical Spatial Structure in 2020 | | | Aggregate |
|---|---|---|---|---|---|
| | | Urban Core | Urban Fringe | Rural Hinterland | |
| Urban core | Impermeable surface | 315.195 | 2.232 | 0 | 317.427 |
| | Grassland | 0.296 | 0.001 | 0 | 0.296 |
| | Cultivated land | 86.756 | 1.881 | 0 | 88.637 |
| | Wasteland | 0.028 | 0 | 0 | 0.028 |
| | Woodland | 1.649 | 0.047 | 0 | 1.696 |
| | Water column | 74.496 | 0.833 | 0 | 75.330 |
| | Aggregate | 478.420 | 4.994 | 0 | 483.415 |
| Urban fringe | Impermeable surface | 86.343 | 168.257 | 0.269 | 254.869 |
| | Grassland | 0.214 | 0.711 | 0.002 | 0.927 |
| | Cultivated land | 121.867 | 280.554 | 1.695 | 404.115 |
| | Wasteland | 0.011 | 0.079 | 0 | 0.090 |
| | Woodland | 3.843 | 14.997 | 0 | 18.840 |
| | Water column | 50.544 | 134.649 | 0.212 | 185.404 |
| | Aggregate | 262.820 | 599.247 | 2.178 | 864.245 |
| Rural hinterland | Impermeable surface | 0.312 | 80.742 | 205.185 | 286.239 |
| | Grassland | 0 | 0.544 | 2.208 | 2.752 |
| | Cultivated land | 1.555 | 422.946 | 5011.656 | 5436.158 |
| | Shrub | 0 | 0 | 0.016 | 0.016 |
| | Wasteland | 0 | 0.014 | 0.128 | 0.142 |
| | Woodland | 0.555 | 15.582 | 480.331 | 496.468 |
| | Water column | 0.253 | 94.949 | 907.433 | 1002.635 |
| | Aggregate | 2.675 | 614.779 | 6606.957 | 7224.411 |
| Aggregate | | 743.916 | 1219.020 | 6609.134 | 8572.071 |

In recent years, Wuhan City has developed rapidly. From the perspective of land-use type transfer, Wuhan occupies more agricultural land in the process of development. From the perspective of territorial spatial structure, urban fringe and rural hinterland are constantly transforming to higher levels and expanding in a circular pattern. Land circulation mainly follows the process of transformation from rural hinterland to urban fringe and from urban fringe to the urban core, and rural hinterland mainly experiences the process of urban fringe to the urban core. The formation of urban land patterns is a long-term process. It is very important to attach importance to the urban fringe and guide the good development of urban construction land.

*4.4. Landscape Pattern Index Changes*

In this paper, the dynamic change of landscape patterns in the fringe area in the past 20 years is analyzed from two aspects: type landscape pattern index and horizontal landscape pattern index, and the development and trend of the fringe area in this period are discussed.

4.4.1. Type Level Landscape Pattern Index Analysis

(1) Area-edge indicator.

The patch area (CA) index represents the area of a certain patch type, and its value is an important basis for determining the dominant landscape in the urban fringe area of Wuhan, and its increment also represents the degree of change in the area of a certain type of landscape patch over a certain period of time [16]. From Table 14, it can be seen that the sequence of CA in the urban fringe area of Wuhan City from 2000 to 2020 is from large to small: agricultural land > impervious surface > water body > woodland > grassland > wasteland. The CA of impervious surface increases from 12,905.19 km$^2$ in 2000 to 39,632.67 km$^2$ in 2020, the largest increase is 14,123.88 km$^2$ during 2010–2020, the increase is the same as that of 2000–2010, which indicates that impervious surface is expanding year by year and shows a stable development. The CA of agricultural land has the largest proportion in all three urban fringe areas, increasing from 14,902.93 km$^2$ in 2000 to 59,048.46 km$^2$ in 2020, which is a large and steady increase; the CA of water bodies also shows a trend of annual increase, increasing by 6640.47 km$^2$ in 2000–2010, but falling back to 1289 km$^2$ in 2010. The CA of water bodies is also increasing year by year, increasing by 6640.47 km$^2$ from 2000 to 2010, but decreasing to 1289.7 km$^2$ between 2010 and 2020. The combination shows that the third urban fringe area of Wuhan mainly has agricultural land and impervious surface as the dominant patches, and the overall situation shows steady growth.

The average plaque size (AREA_MN) index represents an average condition and can be used in combination with plaque density (PD) to compare the differences between plaque types and the degree of aggregation and fragmentation of the plaque types; relatively speaking, the smaller the AREA_MN, the greater the degree of fragmentation. If the plaque density is large and the average area is small, it indicates that the plaques are highly fragmented. If the plaque density is large and the average area is also large, it indicates that the plaques are more uniformly distributed. If the plaque density is small and the average area is large, it indicates that the plaques are concentrated in clusters [16]. From Table 14, the sequence of AREA_MN from large to small in 2000 and 2020 is farmland > water bodies > impervious surface > woodland > grassland > moorland, and the sequence of AREA_MN from large to small in 2010 is farmland > impervious surface > water bodies > woodland > moorland > grassland. The AREA_MN of impervious surface in 2000 increased from 6.6659 to 8.7586 in 2020, which indicates that the patch size of impervious surface is gradually increasing and the fragmentation of patches is decreasing. The AREA_MN of farmland is decreasing year by year, shrinking from 11.7182 in 2000 to 10.6875 in 2020, but farmland is still the largest landscape type in AREA_MN from 2000 to 2020. AREA_MN of woodland is also increasing year by year, from 2.3528 in 2000 to 6.9243 in 2020; AREA_MN of water bodies, grassland, and wasteland has little change. In a comprehensive view, the average

patch area of farmland, impervious surface, and water bodies is large and the patch density is also large, indicating that the distribution of farmland, impervious surface, and water body patches in the Wuhan III urban fringe area is more balanced. The average patch area of woodland is large and the patch density is small, indicating that the woodland is clustered and concentrated in the Wuhan III urban fringe area. While the average patch area of grassland and wasteland is small and the patch density is also small, indicating that the grassland and wastelands are smaller on average and scattered in distribution. In the past two decades, the average size of all patches, except for wasteland and farmland, has tended to increase, indicating that the distribution of land within the Wuhan III urban fringe area is in clusters and blocks, with a trend of increasing concentration.

**Table 14.** Land-use landscape type-level index for urban fringe in Wuhan, 2000–2020.

| Name of Land Type | Year | CA (km²) | NP | PD (Pcs/km²) | LPI (%) | AREA_MN | PAFRAC |
|---|---|---|---|---|---|---|---|
| Impermeable surface | 2000 | 12,905.19 | 1936 | 1.8509 | 3.1461 | 6.6659 | 1.4357 |
| | 2010 | 25,508.79 | 3086 | 1.1718 | 1.8743 | 8.266 | 1.4441 |
| | 2020 | 39,632.67 | 4525 | 1.1939 | 1.4342 | 8.7586 | 1.433 |
| Cultivated land | 2000 | 21,936.51 | 1872 | 1.7897 | 4.1573 | 11.7182 | 1.4367 |
| | 2010 | 40,449.15 | 3914 | 1.4863 | 2.2681 | 10.3345 | 1.4242 |
| | 2020 | 59,048.46 | 5525 | 1.4577 | 2.1769 | 10.6875 | 1.4264 |
| Water column | 2000 | 11,929.05 | 1642 | 1.5698 | 3.1086 | 7.265 | 1.3586 |
| | 2010 | 18,560.52 | 2264 | 0.8597 | 1.2571 | 8.1981 | 1.5458 |
| | 2020 | 19,850.22 | 2068 | 0.5456 | 0.8758 | 9.5988 | 1.3248 |
| Grassland | 2000 | 2.07 | 12 | 0.0115 | 0.0004 | 0.1725 | 1.4951 |
| | 2010 | 93.51 | 220 | 0.0835 | 0.0022 | 0.425 | 1.3858 |
| | 2020 | 73.53 | 164 | 0.0433 | 0.0016 | 0.4484 | 1.4634 |
| Wasteland | 2000 | 0.09 | 1 | 0.001 | 0.0001 | 0.09 | ---- |
| | 2010 | 9.18 | 20 | 0.0076 | 0.0008 | 0.459 | 1.3577 |
| | 2020 | 20.7 | 54 | 0.0142 | 0.0005 | 0.3833 | 1.4271 |
| Woodland | 2000 | 663.48 | 282 | 0.2696 | 0.0977 | 2.3528 | 1.3562 |
| | 2010 | 1884.96 | 314 | 0.1192 | 0.3148 | 6.0031 | 1.3201 |
| | 2020 | 3379.05 | 488 | 0.1288 | 0.2083 | 6.9243 | 1.2784 |

Note: PAFRAC for 2000 wasteland could not be calculated due to the small size of the 2000 wasteland and the 8 cells' neighborhood rule of operation [33].

(2) Dispersion indicators.

Patch number index (NP) indicates the number of patches, or patches of a certain type of landscape, whose value varies from large to small and has a good positive correlation with landscape fragmentation. The NP of the urban fringe data from 2010 to 2020 are: agricultural land > impervious surface > water bodies > woodland > grassland > wasteland. The NP in impermeable surfaces increased from 1936 in 2000 to 4525 in 2020. the NP in cultivated land increased from 1872 in 2000 to 5525 in 2020, the largest proportion. The NP in water bodies and woodlands also increased, but the change was relatively stable, with grassland and wasteland NP remaining largely stable. The increase in NP is a normal trend, given the increase in the size of Wuhan's urban fringe between 2000 and 2020.

The plaque density (PD) index represents the density of certain plaques in a landscape, reflecting the heterogeneity and fragmentation of the landscape as a whole, the fragmentation of particular types of landscape, and the heterogeneity of landscape per unit area. In 2000, from high to low: impermeable surface > farmland > water body > woodland > meadow > moor, and from 2010 to 2020, from high to low: farmland > impermeable surface > water body > woodland > meadow > moor. The PD of impervious surface fell from 1.8509 nr in 2000 to 1.1718 nr in 2010 then rebounded to 1.1939 nr in 2020, with a decreasing trend in overall PD; the PD of farmland had the largest proportion in 2010, but showed an overall decreasing trend; the PD of water bodies, grassland, moorland, and woodland

also decreased year by year. Combined with the number of patches (NP), it can be found that the NP of impervious surface, farmland, and water bodies in Wuhan increased year by year from 2000 to 2020, but the PD decreased year by year, indicating that although the number of patches was increasing with the increase of the absolute area of the urban fringe area, the patches gradually showed a trend of concentration.

The maximum patch count (LPI) index can help us to identify the dominant landscape elements in the Wuhan III urban fringe area. The 2000–2020 LPI large to small sequence is: farmland > impervious surface > water body > woodland > grassland > wasteland, as can be seen from Table 14, the LPI index of impervious surface shrinks from 3.1461% in 2000 to 1.4342% in 2020, and the LPI of farmland decreases from 4.1573% in 2000 to 2.1769% in 2020. Cropland the LPI declined from 4.1573% in 2000 to 2.1769% in 2020, an overall downward trend, but it is still the dominant landscape in the urban fringe of Wuhan City in the third phase. Water LPI declined from 3.1086% in 2000 to 0.8758% in 2020, a significant decrease. The remaining landscape types showed little change in LPI. The results show that the LPI of farmland and the impermeable surface is generally declining, but it is still the dominant landscape element in the urban fringe of Wuhan and contributes to the whole landscape.

(3) Shape indicators.

The perimeter fractional dimensional index (PAFRAC), which reflects a certain extent the degree of human interference in the landscape pattern, is smaller when the index value tends to 1, indicating that if the patch shape in the landscape is simpler, then it is likely to be less interfered by human activities, while the index value is larger, i.e., close to 2, the more complex, the more disturbed by human activities [16]. PAFRAC in 2000 is in descending order: grassland > agricultural land > impervious surface > water bodies > woodland PAFRAC in 2010 is in descending order: water bodies > impervious surface > agricultural land > grassland > moorland > woodland PAFRAC in 2020 is in descending order: grassland > impervious surface > badlands > farmland > waterbodies > woodland. The patches with relatively large changes in PAFRAC in the three periods of data were water bodies and heathland. The PAFRAC for water bodies increased from 1.3586 in 2000 to 1.5458 in 2010, then decreased to 1.3248 in 2020, with the overall shape of the water bodies being significantly altered. On the whole, although the shape index changes slightly, it shows a more stable state overall. It indicates that the means of development and utilization are more stable in the development and construction of various types of land in the urban fringe area.

### 4.4.2. Horizontal Hierarchical Landscape Pattern Index Analysis

According to the horizontal landscape pattern index, patch density, Shannon diversity and Shannon evenness of Wuhan City's urban fringe area generally declined from 2000 to 2020. The data from 5.4952/km$^2$, 1.1959, and 0.6146 in 2000 to 3.384 units/km$^2$, 0.9879, 0.5077 indicate the fragmentation of the landscape in the third urban fringe area of Wuhan is on a decreasing trend. The landscape is gradually distributed in clusters, the landscape heterogeneity continues to decrease during the 20 years. The performance of landscape types gradually shows a homogeneous character, and the landscape is dominated by two dominant patches of impervious surface and agricultural land. In Wuhan, the landscape shape index and aggregation Index increased from 32.8363 and 94.1677 in 2000 to 42.4062 and 96.0006 in 2020 (Table 15), respectively, reflecting a marked irregular trend in landscape patches morphology in three urban fringe areas of Wuhan. The decrease of landscape dispersion, with the landscapes clustered and blocked together. Between 2000 and 2020, new construction sites have not exacerbated or even reduced landscape fragmentation overall, partly due to erosion of non-dominant landscape patches, resulting in reduced landscape heterogeneity and homogenization of landscape types.

**Table 15.** Land-use landscape level index for urban fringe areas in Wuhan, 2000–2020.

| Year | PD | LSI | SHDI | SHEI | AI |
|------|------|---------|--------|--------|---------|
| 2000 | 5.4952 | 32.8363 | 1.1959 | 0.6146 | 94.1677 |
| 2010 | 3.7297 | 35.9494 | 1.0068 | 0.5174 | 95.9574 |
| 2020 | 3.384 | 42.4062 | 0.9879 | 0.5077 | 96.0006 |

## 5. Discussion

### 5.1. Integration of Urban Fringe Dynamics with Urban Spatial Extension

5.1.1. The Urban Fringe Expansion Pattern of Wuhan

Population growth, economic development, political factors, the urbanization process, and other socio-economic factors play a dominant role in urban spatial expansion [34,35]. Economic growth is the primary factor influencing the spatial expansion of urban fringe areas. The industrial structure is developing in the direction of advanced, and the proportion of industrial and tertiary industries is gradually increasing, which also increases the demand for industrial land and stimulates the rapid outward expansion of the city. At the same time, industrial growth has led to the concentration of population in the cities, resulting in the expansion of urban construction and living space, accelerating the process of regional urbanization, and thus pushing the boundaries of urban fringe areas to expand outward. Economic development provides strong financial support for the construction of urban infrastructure, and the infrastructure itself requires a large amount of land, which further leads to the expansion of urban fringe areas. As can be seen in Figure 7, Wuhan's radial highways play an important role as urban expansion axes, with the urban fringe growing radially along the transportation corridors. The direction of spatial expansion is guided and determined by urban planning. The development direction of the three towns' land use is clearly stated in the Wuhan Master Plan: Wuchang is to be developed to the east and south, toward Nanhu, Guanshan, and Baishazhou. Hankou is to be developed and built around Zhaojiaqiao, while other land use is to be strictly controlled (land for the tertiary industry is reserved for development). Hanyang is to be developed moderately toward Shengguandu, Qinhuangkou, and the area north of Inkanghu. Qingshan is to be developed appropriately to the south (Wudong, Baiyu Mountain, etc.). The construction of the Nanhu community began, becoming the most southern residential area in Wuchang. During 2004–2005, various urban industrial parks and suburban industrial parks began to be built vigorously, and industrial land in the fringe area grew particularly rapidly. In 2009, the Wuhan railway station was built and opened to traffic, marking the northeastern extension of Wuhan's railway transportation system and promoting the construction of the Qingshan and Hongshan fringe areas [36].

Natural geography is an important basic condition of urban space expansion, which directly influences the potential, direction, and speed of urban space expansion. According to the overall morphological changes of the Wuhan fringe area, Wuhan is located in the middle of the Jianghan Plain, the topography is mainly plain, the remaining hills are scattered from east to west in the middle, and there are many lakes and ponds in the city. In the Hankou area of Wuhan, the low topography and severe waterlogging in the rainy season north of the Zhanggong dike limit the outward expansion of the fringe area to a certain extent. In the Wuchang area, due to the strong influence of large natural water bodies (such as the East Lake), the fringe area has limited space for development and is mainly developed in strips between waters; in the Hanyang area, due to the topographic constraints of lakes and hills, the fringe area jumps and extends outward along traffic routes. The Yangtze River and Hanshui cross Wuhan, and the fringe of the city continues to expand, also showing a tendency to expand along the shores of the two rivers.

In the past 20 years, the transformation of Wuhan's urban spatial structure has been mainly from the urban fringe to the urban core, and from the rural hinterland to the urban fringe, with the spatial structure showing a clear "urban core-urban fringe-rural hinterland" circle. The dynamic attitude of different land types within the urban fringe area varies

significantly, and a large amount of agricultural land and water bodies are transformed into impervious surfaces. Most of the rural hinterland has gone through a period of urban fringe before being transformed into an urban core.

### 5.1.2. Wuhan Marginal Zone Expansion vs. Beijing, Guangzhou, and Changchun

Wuhan, Beijing, Guangzhou, and Changchun are located in central China, northern China, southern China, and northeastern China, respectively. The cities differ widely and can be compared as representative cities in different regions (Table 16).

**Table 16.** Comparison of Urban Expansion Characteristics of Wuhan, Beijing, Guangzhou, and Changchun.

| | Wuhan | Beijing | Guangzhou | Changchun |
|---|---|---|---|---|
| Expansion speed | Urban core expansion rate of 23.03 km$^2$/a and urban fringe rate of 39.06 km$^2$/a in 2000–2010; urban core expansion rate of 26.07 km$^2$/a and urban fringe rate of 35.51 km$^2$/a in 2010–2020 | Urban core expansion rate of 18.58 km$^2$/a and urban fringe rate of 57.24 km$^2$/a in 1994–1999 1999–2004 Urban core expansion rate 9.36 km$^2$/a and urban fringe expansion rate 195.32 km$^2$/a | From 1990–2000, the area of the urban core increased by 27.86 km$^2$/a per year and the area of the urban fringe grew by 9.6 km$^2$/a per year Annual growth rate of urban core area 9.4 km$^2$/a and urban fringe area −2.05 km$^2$/a from 2000–2009 | Expansion rates of 15.41 km$^2$/a in the urban core and 50.33 km$^2$/a in the urban fringe, 1995–2005; 10.42 km$^2$/a in the urban core and 2.98 km$^2$/a in the urban fringe, 2005–2015. |
| Direction of expansion | 2000–2020 mainly in five directions: new southwest, west, north, northeast, east; 2000–2010 in south, west, northeast; 2010–2020 in southwest, north, east | 1994–1999 urban fringe expansion mainly to the northeast of the city, urban core expansion mainly to the southwest; 1999–2004 tends to even out in all directions | From 1990 to 2000, the strongest expansion of the urban core was located in the northern part of Guangzhou, and the urban fringe areas were in the southern and northern parts of Guangzhou; from 2000 to 2009, the urban core expanded mainly in the southern and eastern parts, and the urban fringe areas were concentrated in Panyu | Urban fringe area extends mainly to the east, southeast and north from 1995–2015; mainly to the east, southeast and south from 1995–2005; mainly to the north from 2005–2015 |
| Expansion model | 2000–2010 urban fringe is mainly outward expansion; 2010–2020 urban fringe is mainly outward expansion and internal infill | 1994–1999 focused mainly on infill and development within the marginal zone; 1999–2004 focused mainly on outward expansion | Outward expansion in urban fringe areas 1990–2000, in-fill 2000–2009 | Urban fringe areas were mainly outward sprawling from 1995–2015; mainly outward sprawling from 1995–2005; and mainly in-fill from 2005–2015 |
| Expansion area | The urban fringe expansion areas are mainly in Hongshan District, Caidian District, East-West Lake District, Hannan District, Huangpi District, Jiangxia District and Xinzhou District; the core expansion areas are in Jiangan District, Qiaokou District, Jianghan District, Wuchang District, Qingshan District, Hongshan District, Caidian District, East-West Lake District, and Jiangxia District | Districts with large urban fringe expansion areas Tongzhou, Changping, Daxing, Shunyi | The main expansion areas of the urban core are Panyu, Baiyun, and Luogang districts; the proportion of urban fringe areas in Baiyun, Luogang, Huadu, and Nansha districts increased and then decreased during the period 1990–2009 | - |
| Landscape pattern | The fragmentation of the landscape in Wuhan's three urban fringe areas is on a declining trend, and landscape heterogeneity has continued to decrease over 20 years | - | Increasing fragmentation in urban core and fringe areas from 1990–2000 and decreasing fragmentation from 2000–2009 | The number of patches in the urban fringe increased and the average patch size decreased from 1995–2005. From 2005–2015, the average patch size increased |

In terms of the number of expansions, each city has gone through different stages of expansion, with different rates of expansion in different stages; this is not a good idea to compare them cross-sectionally because of the different years of study in each city; and there is no simple law to express the development pattern of each city because of the large differences in economic volume and different stages of development, so it is clear that the factors affecting the expansion of each city fringe are very complex and therefore deriving different characteristics.

In terms of expansion direction and expansion area, different cities also show different characteristics in different years. One of the most direct factors is the policy and planning of the city [37]. A deeper reason may be that geography and the influence of surrounding cities show great strength. There is more water to the southeast of Wuhan, more mountains to the north of Beijing, and more mountains to the east of Changchun, all of which limit the development in these areas [24,38]. While the strong economic volume of Shenzhen and Hong Kong to the northeast of Guangzhou attracts the city to develop in the direction of these two cities. In addition, different kinds of literature have also mentioned that transportation arteries are one of the main factors affecting expansion [24,37,38].

In terms of expansion patterns, with the exception of Beijing, different cities show a shift from sprawl to internal infill, which is more in line with the current strict land management regime. Data for Beijing do not support this conclusion for the time being, but since their data are for the year up to 2004, they do not reflect the latest national policies.

The landscape pattern fragmentation fringe area of Wuhan, Guangzhou, Changchun, and so on shows that the urban development trend is gradually shifting from extensive to orderly utilization, the city has a clear tendency to expand, up to a rounded shape, and this development trend is beneficial to the overall formation of the landscape in a small scale; but on a large scale, however, it tends to form a pattern of spreading the pie.

### 5.2. Landscape Ecological Service Function of Landscape Patterns in Urban Fringe Areas

Landscape ecological service function is the most direct and important role of urban fringe in urban development. As the hinterland around the city, the landscape ecological function of the urban fringe is the most important supplement and enhancement to urban function. With Wuhan's rapid urbanization and human transformation of nature, the natural landscape patches in Wuhan's fringe areas have gradually transformed into artificial landscape patches in the past 20 years. With obvious conversion within the patches, destruction of the original natural ecosystem, and pollution of the environment, breaking the balance of the natural ecological effects of self-circulation flow, and although the rate of destruction has slowed down in recent years, the urban fringe areas are still facing serious problems. From 2000 to 2010, the fragmentation of impervious surfaces, water bodies, and woodlands increased rapidly. Wuhan entered a period of rapid development, the city continued to expand into the rural hinterland, and a large amount of farmland and water bodies were transformed into impervious surfaces, resulting in a complex landscape structure [39]. In general, the landscape patches in the urban fringe area of Wuhan City tend to become more irregular in shape, the landscape dispersion continues to decrease, the trend of fragmentation weakens, the landscape is gradually distributed in clusters and blocks, and the landscape type performance gradually shows the characteristics of homogenization. The plaque density decreased and the overall distribution is concentrated. It can be seen that the difference in physical properties of landscape pattern in Wuhan City leads to changes in system attributes such as surface reflectance, which affects ecosystem service processes in Wuhan's city center [40,41]. Supply services are weaker and management and supporting services are stronger when the anthropogenic disturbance is low. At moderate levels of disturbance, provisioning services are stronger and regulating and supporting services are weaker. Landscape degradation under high disturbance threatens the provision of multiple ecosystem services in central Wuhan.

### 5.3. Policy Recommendations

There is a regrouping of various land types after severe division, in line with the trend of urban expansion. However, with the continuous expansion of Wuhan City, the land resources and ecological environment of the urban fringe are inevitably under considerable pressure. Land-use landscape types are increasingly affected by human activities. In order to promote the healthy and orderly development of landscape ecological function and the harmonious development of Wuhan society, economy, and ecology, the following suggestions are put forward.

(1) From the perspective of different types of cities, urban development is constrained and guided by policies and planning, which is also the result of our strict land and planning management systems. Therefore, urban planning needs to provide scientific guidance for urban development. Among them is the role of traffic arteries in promoting urban expansion, which is reflected in different cities, and conversely, the crossing of traffic arteries should be reduced for ecological areas so that the ecological area can be preserved. The Wuhan City Territorial Spatial Master Plan (2021–2035), released in Wuhan, sets aside six green wedges for the city, which, in addition to strictly limiting the development of construction land, should also reduce the crossing of traffic arteries in them.

(2) Under the strict land management system, the development mode of large-scale outward expansion will be difficult and will be gradually replaced by the urban expansion mode of internal filling. This shows that the expansion of the urban fringe area has slowed down, instead, the interior of the urban fringe area has been continuously transformed into the urban core area. The new general plan of Wuhan City proposes the overall spatial form of a green wedge, so the future development should strengthen the internal de-elopement of the existing urban area, and at the same time strictly control the encroachment of green space by the land around the green wedge.

(3) Incorporate the idea of landscape ecological protection. In order to maximize the ecological effect of the urban fringe area and reduce the pressure on the ecological environment of the urban fringe area, balance the relationship between development and ecology, and ensure the sustainable development of urban construction in Wuhan.

(4) Enhance policy concern for urban fringe areas. With the development of the economy and society, the city keeps expanding outward and encroaching on arable land, water bodies, and other land, the protection and treatment of land with ecological regulation function, such as agricultural land and ecological land, should be increased to improve the ecological benefits of the landscape.

## 6. Conclusions

After the previous analysis and discussion, the contributions of this study are: Firstly, the quantitative extraction of Wuhan urban fringe areas from two aspects of impervious surface ratio and landscape flocculation and their correction using population density threshold to delineate the relatively accurate Wuhan fringe areas. Secondly, the dynamic attitude analysis and land-use transfer matrix analysis of land use in Wuhan to quantify the 2000–2020 Wuhan City's spatial structure and land-use spatio-temporal dynamic evolution process, analyze the coupling relationship between the dynamic changes of Wuhan City fringe area and urban spatial expansion from 2000–2020. Analyze the characteristics and patterns of Wuhan City fringe area expansion and conduct analogous analysis with other cities. Third, analyze the type-level landscape pattern index and horizontal-level landscape. The dynamic evolution of the Wuhan urban fringe landscape pattern in the past 20 years is analyzed from two aspects, namely the type-level landscape pattern index and the horizontal-level landscape pattern index, and the landscape ecological services provided by the Wuhan urban fringe to the city are analyzed through the dynamic change process of the Wuhan fringe landscape pattern presented by the landscape pattern index at different time periods. Combining with the concept of "green wedge" in the spatial planning of Wuhan City and the construction of landscape pattern of the fringe area, the development suggestions are put forward.

This study also has shortcomings for further improvement: (1) Urban fringe areas are highly ambiguous and dynamic, and there are different approaches to extract urban fringe areas in the existing literature, including spatial clustering method, breakpoint method, population density gradient analysis method, etc. In the future, multiple data, multiple methods, and multiple resolutions can be used to further improve the accuracy of urban fringe area delineation. (2) Wuhan is known as the "city of 100 lakes" due to its large number of lakes. However, when the moving *t*-test method is used to obtain the mutation points of impermeability ratio of surface water, pseudo-mutation points appear on the

surface of the water and need manual removal, which increases the subjectivity of the study. (3) The driving forces of spatial evolution and landscape pattern evolution in Wuhan fringe areas have not been analyzed and need to be further improved in subsequent studies.

**Author Contributions:** Conceptualization, Yan Long and Xuejun Liu; methodology, Yan Long and Xuejun Liu; software, Shiqi Luo, Xi Liu and Tianyue Luo; validation, Yan Long and Xuejun Liu; formal analysis, Yan Long and Xuejun Liu; investigation, Shiqi Luo and Xi Liu resources, Yan Long and Xuejun Liu; data curation, Xuejun Liu; writing—original draft preparation, Yan Long, Shiqi Luo and Xi Liu; writing—review and editing, Yan Long, Shiqi Luo, Xi Liu and Xuejun Liu; visualization, Shiqi Luo and Xi Liu; super-vision, Yan Long and Xuejun Liu; project administration, Yan Long; funding acquisition, Yan Long All authors have read and agreed to the published version of the manuscript.

**Funding:** This research was funded by A Philosophy and Social Science Research Project under Hubei Higher Education of 2021 (21Y029).

**Institutional Review Board Statement:** Not applicable.

**Informed Consent Statement:** Not applicable.

**Data Availability Statement:** https://doi.org/10.5281/zenodo.7068626 (accessed on 5 June 2022).

**Conflicts of Interest:** The authors declare no conflict of interest.

## Appendix A

**Table A1.** Wuhan Land Use Transfer Matrix of urban core area and urban fringe, 2000–2010.

| Urban Territorial Structure | Land Types | 2010 Urban Core Area | | | | | | | 2010 Urban Fringe Area | | | | | | |
|---|---|---|---|---|---|---|---|---|---|---|---|---|---|---|---|
| | | Impervious Surface | Grassland | Farmland | Unused Land | Forest Land | Water Body | Total | Impervious Surface | Grassland | Farmland | Unused Land | Forest Land | Water Body | Total |
| 2000 urban core area | Impervious surface | 168.313 | 0 | 0.004 | 0 | 0 | 0.189 | 168.506 | 0.541 | 0 | 0 | 0 | 0 | 0.003 | 0.544 |
| | Grassland | 0.007 | 0.001 | 0 | 0 | 0 | 0 | 0.008 | 0 | 0 | 0 | 0 | 0 | 0 | 0 |
| | Farmland | 20.182 | 0.019 | 16.155 | 0.001 | 0.058 | 0.479 | 36.895 | 0.395 | 0 | 0.526 | 0 | 0 | 0.027 | 0.948 |
| | Unused land | 0.001 | 0 | 0 | 0.004 | 0 | 0 | 0.005 | 0 | 0 | 0 | 0 | 0 | 0 | 0 |
| | Forest land | 0.047 | 0.001 | 0.214 | 0 | 0.712 | 0.003 | 0.976 | 0 | 0 | 0.001 | 0 | 0.002 | 0 | 0.003 |
| | Water body | 5.961 | 0.001 | 2.831 | 0.002 | 0 | 36.231 | 45.025 | 0.125 | 0 | 0.147 | 0 | 0 | 0.170 | 0.443 |
| | Total | 194.511 | 0.022 | 19.205 | 0.007 | 0.769 | 36.902 | 251.416 | 1.062 | 0 | 0.674 | 0 | 0.002 | 0.201 | 1.938 |
| 2000 urban fringe | Impervious surface | 62.163 | 0 | 0.010 | 0 | 0 | 0.136 | 62.309 | 66.312 | 0 | 0.056 | 0 | 0 | 0.291 | 66.659 |
| | Grassland | 0.003 | 0.002 | 0 | 0.001 | 0 | 0.002 | 0.007 | 0.010 | 0.002 | 0.002 | 0 | 0 | 0 | 0.013 |
| | Farmland | 39.705 | 0.221 | 47.676 | 0.007 | 0.077 | 1.190 | 88.876 | 41.913 | 0.157 | 83.802 | 0.003 | 0.701 | 3.723 | 130.298 |
| | Unused land | 0 | 0 | 0 | 0.001 | 0 | 0 | 0.001 | 0 | 0 | 0 | 0 | 0 | 0 | 0 |
| | Forest land | 0.042 | 0 | 0.296 | 0 | 0.812 | 0.004 | 1.154 | 0.061 | 0 | 0.609 | 0 | 4.781 | 0.008 | 5.459 |
| | Water body | 5.411 | 0.024 | 6.763 | 0.012 | 0 | 29.815 | 42.024 | 7.682 | 0.018 | 9.667 | 0.016 | 0.012 | 59.758 | 77.153 |
| | Total | 107.323 | 0.247 | 54.745 | 0.021 | 0.889 | 31.146 | 194.371 | 115.979 | 0.177 | 94.135 | 0.019 | 5.494 | 63.779 | 279.583 |

**Table A2.** Wuhan Land Use Transfer Matrix of rural heartland, 2000–2010.

| Urban Territorial Structure | Land Types | 2010 Urban Fringe Area Area | | | | | | | Rural Heartland 2010 | | | | | | |
|---|---|---|---|---|---|---|---|---|---|---|---|---|---|---|---|
| | | Impervious Surface | Grassland | Farmland | Unused Land | Forest Land | Water Body | Total | Impervious Surface | Grassland | Farmland | Shrub | Unused Land | Forest Land | Water Body |
| 2000 rural heartland | Impervious surface | 48.334 | 0 | 0.046 | 0 | 0 | 0.341 | 48.721 | 167.094 | 0.003 | 0.222 | 0 | 0 | 0.008 | 5.959 |
| | Grassland | 0.114 | 0.028 | 0.026 | 0 | 0 | 0.002 | 0.17 | 0.491 | 0.835 | 0.863 | 0 | 0.004 | 0.414 | 0.328 |
| | Farmland | 83.561 | 0.711 | 281.746 | 0.006 | 1.036 | 10.587 | 377.647 | 106.612 | 1.349 | 5248.956 | 0 | 0.008 | 57.31 | 167.578 |
| | Shrub | 0 | 0 | 0 | 0 | 0 | 0 | 0 | 0 | 0 | 0.004 | 0.016 | 0 | 0.008 | 0 |
| | Unused land | 0 | 0 | 0 | 0 | 0 | 0 | 0 | 0.308 | 0.143 | 0.087 | 0 | 0.118 | 0 | 0.385 |
| | Forest land | 0.232 | 0.001 | 1.41 | 0 | 12.308 | 0.001 | 13.951 | 1.803 | 0.371 | 46.904 | 0 | 0 | 438.567 | 0.427 |
| | Water Body | 5.588 | 0.01 | 26.078 | 0.065 | 0.001 | 110.492 | 142.234 | 9.931 | 0.052 | 139.122 | 0 | 0.013 | 0.161 | 827.957 |
| | Total | 137.829 | 0.75 | 309.306 | 0.071 | 13.345 | 121.423 | 582.723 | 286.239 | 2.753 | 5436.158 | 0.016 | 0.143 | 496.468 | 1002.634 |

**Table A3.** Wuhan Land Use Transfer Matrix of urban core area and urban fringe, 2010–2020.

| Urban Territorial Structure | Land Types | 2020 Urban Core Area | | | | | | | 2020 Urban Fringe Area | | | | | | |
|---|---|---|---|---|---|---|---|---|---|---|---|---|---|---|---|
| | | Impervious Surface | Grassland | Farmland | Unused Land | Forest Land | Water Body | Total | Impervious Surface | Grassland | Farmland | Unused Land | Forest Land | Water Body | Total |
| 2010 urban core area | Impervious surface | 313.699 | 0 | 0.060 | 0 | 0 | 1.436 | 315.195 | 2.218 | 0 | 0 | 0 | 0 | 0.014 | 2.232 |
| | Grassland | 0.269 | 0.007 | 0.009 | 0.011 | 0 | 0 | 0.296 | 0.001 | 0 | 0 | 0 | 0 | 0 | 0.001 |
| | Farmland | 32.521 | 0.019 | 51.831 | 0.024 | 0.352 | 2.008 | 86.756 | 0.285 | 0 | 1.582 | 0 | 0.012 | 0.002 | 1.881 |
| | Unused land | 0.004 | 0 | 0 | 0.019 | | 0.004 | 0.028 | 0 | 0 | 0 | 0 | 0 | | 0 |
| | Forest land | 0.011 | 0 | 0.209 | 0 | 1.430 | 0 | 1.649 | 0 | 0 | 0.007 | 0 | 0.040 | 0 | 0.047 |
| | Water body | 2.175 | 0 | 4.518 | 0 | 0.001 | 67.803 | 74.496 | 0.017 | 0 | 0.037 | 0 | 0 | 0.779 | 0.833 |
| | Total | 348.679 | 0.026 | 56.627 | 0.054 | 1.783 | 71.252 | 478.420 | 2.521 | 0 | 1.626 | 0 | 0.052 | 0.795 | 4.994 |
| 2010 urban fringe area | Impervious surface | 86.139 | 0 | 0.020 | 0 | 0 | 0.184 | 86.343 | 166.537 | 0.004 | 0.478 | 0 | 0 | 1.239 | 168.257 |
| | Grassland | 0.192 | 0.002 | 0.018 | 0.002 | | | 0.214 | 0.559 | 0.031 | 0.087 | 0.031 | 0.002 | 0 | 0.711 |
| | Farmland | 38.574 | 0.067 | 80.745 | 0.037 | 0.613 | 1.831 | 121.867 | 59.332 | 0.147 | 215.642 | 0.014 | 1.458 | 3.960 | 280.554 |
| | Unused land | 0.001 | 0 | 0 | 0.010 | 0 | 0 | 0.011 | 0.056 | 0 | 0 | 0.023 | 0 | 0 | 0.079 |
| | Forest land | 0.034 | 0 | 0.485 | 0 | 3.323 | 0 | 3.843 | 0.085 | 0 | 1.222 | 0 | 13.687 | 0.003 | 14.997 |
| | Water body | 1.824 | 0 | 6.018 | 0.002 | 0 | 42.700 | 50.544 | 4.894 | 0.007 | 13.673 | 0.015 | 0.004 | 116.056 | 134.649 |
| | Total | 126.763 | 0.069 | 87.286 | 0.051 | 3.937 | 44.715 | 262.820 | 231.463 | 0.190 | 231.102 | 0.084 | 15.151 | 121.258 | 599.247 |

**Table A4.** Wuhan Land Use Transfer Matrix of rural heartland, 2010–2020.

| Urban Territorial Structure | Land Types | 2020 Urban Fringe Area | | | | | | | Rural Heartland 2020 | | | | | | | |
|---|---|---|---|---|---|---|---|---|---|---|---|---|---|---|---|---|
| | | Impervious Surface | Grassland | Farmland | Unused Land | Forest Land | Water Body | Total | Impervious Surface | Grassland | Farmland | Shrub | Unused Land | Forest Land | Water Body | Total |
| Rural heartland 2010 | Impervious surface | 79.858 | 0 | 0.097 | 0 | 0 | 0.787 | 80.742 | 198.329 | 0 | 1.073 | 0 | 0 | 0.001 | 5.782 | 205.185 |
| | Grassland | 0.409 | 0.045 | 0.076 | 0.009 | 0.004 | 0.001 | 0.544 | 0.202 | 0.159 | 1.348 | 0 | 0 | 0.368 | 0.131 | 2.208 |
| | Farmland | 77.444 | 0.473 | 333.048 | 0.030 | 4.691 | 7.260 | 422.946 | 101.486 | 0.239 | 4632.126 | 0 | 0.008 | 161.431 | 116.366 | 5011.656 |
| | Shrub | 0 | 0 | 0 | 0 | 0 | 0 | 0 | 0 | 0 | 0.003 | 0.004 | 0 | 0.010 | 0 | 0.016 |
| | Unused land | 0.010 | 0.004 | 0 | 0 | 0 | 0 | 0.014 | 0.012 | 0 | 0.055 | 0 | 0.002 | 0 | 0.059 | 0.128 |
| | Forest land | 0.139 | 0.002 | 1.561 | 0 | 13.879 | 0.001 | 15.582 | 0.506 | 0.009 | 35.487 | 0 | 0 | 444.202 | 0.127 | 480.331 |
| | Water body | 4.146 | 0.020 | 22.447 | 0.082 | 0.001 | 68.254 | 94.949 | 4.067 | 0.008 | 155.687 | 0 | 0.051 | 0.785 | 746.834 | 907.433 |
| | Total | 162.006 | 0.544 | 357.229 | 0.120 | 18.576 | 76.303 | 614.779 | 304.602 | 0.416 | 4825.778 | 0.004 | 0.061 | 606.796 | 869.300 | 6606.957 |

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
