# Peer review of "Research on the Dynamic Evolution of the Landscape Pattern in the Urban Fringe Area of Wuhan from 2000 to 2020"

_ijgi, doi:10.3390/ijgi11090483_

Round 1
Reviewer 1 Report
A very interesting topic to be researched as findings can help various decision-makers when considering the layout of landscape patterns in the urban fringe area. However, there are many revisions requested to make this publishable. The largest revisions include restructuring the introduction and literature review. The literature review needs to be better organized with themes trying to be relayed. I suggest using landscape patterns as an outline for the literature review and then walking through each criterion and the effects of each criterion. There are many helpful references, they just need to be organized better and categorized into themes you are trying to relay in the literature review. The hypotheses should be written based on how the results are presented.
The authors need to justify the study's sampling - why do only urban fringe areas in Wuhan were chosen? The authors also need to justify the background/ and conditions as these may also influence their perception in justifying the validity of the findings. In the section results, it is not clear how the population density has influenced the paper and how helped to achieve its targets. The comparison between findings and previous research should include the parameters for comparison. The author should stipulate the "practical suggestions" implied here. The manuscript needs to be edited for English language grammar, spelling, and verb tense. The main steps for Fragstats should be explained.
Author Response
Dear Professor Reviewer.
We thank you and the reviewers for your thoughtful comments, many of which have been incorporated into the revised manuscript. Please see the attachment for more details

Reviewer 2 Report
The paper entitled Research on the Dynamic Evolution of the Landscape Pattern in the Urban Fringe Area of Wuhan From 2000 to 2020 presents an interesting method for evaluating the landscape pattern evolution for a urban fringe area. Still, there should be clarified a few things that are stated in the paper. Also, the are certain observations that are to be considered, as follows:
Line 1-2: At the beginning of the abstract, it would be better to give a clear definition of the urban fringe expression.
Line 20-22: Some quantitative data would be necessary regarding the urban expansion as a final result of the study. How much of the agricultural land surface or other category has actually been lost?
Line 37: why scholars? – researchers maybe...?
Line 47-48: Consider revision for the entire phrase.
Line 49-67: Still not a clear definition of the urban fringe. We appreciate the examples and explanations, but you should be clearer about the concept, like you did for the landscape pattern in rows 68-70.
Line 95: better use tendency rather than law.
Line 96: better use trend rather than law.
Figure 1: Title for the first map – Republic of China
Title for the third map – Wuhan urban area
Also try a smaller north arrow for the third map (with Wuhan).
Figure 2 – instead of Technical route, use Workflow analysis
Line 116: please provide the exact link of the downloaded data (and accessed date). Also specify the resolution of the landcover data. What Is the source of the landcover data? What is the accuracy of the used data?
Line 151: - landscape flocculation – not sure about this concept. Flocculation is usually used in chemistry. I have never seen it associated with the term landscape. Please find a citation for the concept or use a different translation. If you do propose a new concept, you must come with a strong, citation based, argumentation.
Line 157: Use a citation to strengthen your decision in choosing the two criteria in the final analysis.
Line 191: Please use a citation for the concept. This definition worked better when you first used the term.
Line 227: Citation needed for landscape index.
Line 257-272: Consider revision of the phrase. Too long and difficult to follow. Use the past tense only. Segment the phrase for a better understanding, please.
Line 266 – better explanation of what mutation points are. It almost goes similar to landscape flocculation…
Line 277: please offer more details about what satellite image you have used for the analysis (name of the sensor, year, source etc).
Line 294: which is the source of the land cover data?
Line 302: we suggest using more general accepted terms when it comes to technical procedures. ArcGIS terminology is not well known by all GIS users. Grid area would be an idea instead of fishing net.
Figure 5 – please adapt the legend so would be suitable for reading. No bold for the font, reduce decimals to only two, smaller north arow.
Figure 6 is missing! Please review the manuscript and revise.
Line 353: Consider revising the phrase.
Line 362: Can you replace “a lot” with the exact percentage of farmland and water bodies that were replaced with urban land area?
Table 2 – The table is useful, but unfortunately hard to follow. For a better reading I recommend to change the layout and put in three sections on the vertical side (urban core, urban fringe and hinterland) and on horizontally the years. Thus, one could observe faster the change over the years on each land type. Also consider showing the numbers in percentages too for a more relevant perspective.
Line 370 – instead of greatly use significantly. Also, when writing about evolution and changes, try to put numbers rather than words in order to add more consistency to the argumentation.
Table 6: at PD column, instead of pieces – write nr.
Line 551: write nr. instead of pieces.
Line 555: „The landscape is dominated by two dominant patches of impervious surface and farmland.”
Rephrase: The landscape is dominated by two types of patches, of impervious surface and farmland respectively
Lines 557-564 – Consider revising the phrases. The structure of the phrase makes the meaning a bit unclear.
Line 566 – You should mention the place you refer to (Wuhan) in the first sentence. Rapid should be spelled with a lowercase.
Line 571 – grievously – not sure what the authors meant to say here?
Lines 604-608: Phrase too large. Please break the phrase in smaller sentences, making it easier to follow.
Chapter 5: - Please separate the two chapters. Write discussion separately. The ideas are well presented, but somehow, they lack some order. The authors should consider to rearrange them by answering to some questions regarding their findings: what are the causes and what could be the effects of this expansion (effects should be presented separately on different levels, such as nature, economy, social, architecture etc)? Are there other areas In China or other countries that have experienced something similar? If so, what happened there, and what could be the solutions for this matter? We understand that this is not the main purpose of the paper, but going a bit further with the analysis would only bring more value to the study.
Chapter 6 – Conclusions: in this chapter you should briefly present the conclusions of your study and the impact that it might have upon the study of the matter discussed in the paper.
Please consider revising the English over the entire paper.
Author Response

(The authors gave the same response as above.)

Reviewer 3 Report
The article approaches a very interesting and timely topic. The manuscript's scope fits within the IJGI journal scope and follows a clear structure. It is particularly suitable and detailed the organisation of the "3. Data and Methods" section. However, some comments are given in order to improve and clarify different aspects of the revised manuscript.
In the Introduction, as well as in the Abstract, the terms "landscape flocculation degree" and "landscape disorder population density" are introduced. It would be useful to briefly disambiguate these very specific terms.
Also, it is desiderable to change the expression "our country's" to "China's"(line 32).
There is a mistake in reference [16]. In the text (line 59) appears He Jianhua cited, but in the final list of references the work cited is Gallent et alt. Please check which one is the appropiate.
In Section 3. Data and Methods it would be convenient to include a reference for L H Russwurm research. There is a typo in the sub-section number (line 196).
In "Section 4. Results and Analysis" is clearly commented on the significant data shown in tables and figures. It would be recommended to clarify why are developed "180 concentric circles at the center with a radious difference of 500 meters"(line 259). Is 500 meters the measurement of an specific landscape feature? Additional explanation would help readers to better understand the designed process of analysis.
Please revise figures' numbers, specifically figures 4 (line 285) and 5 (line 313)
In the same Section 4, lines 336-337, the sentence "For the overlapping aera, it can be directly divided into the urban fringe area of Wuham, for the non-overlapping area" is a little cryptic. It is recommended to rephrase the sentence or the paragraph to improve readability.
In Section 5 Conclusion and Discussion section would be desiderable additional references to the previous section results, in order to better link both sections referring back to the analysis development. Also, a final comment may also include some suggestions for improvement or advancing future research directions.
Author Response

(The authors gave the same response as above.)

Reviewer 4 Report
The manuscript is well done and rather clearly written. I have only a very strong concern. What is the true original contribution of this paper to urban literature? In other words, I would remark that we are plenty of papers describing land-use change and urban expansion all over the world. Wuhan in going in the same direction. This is true, nothing bad. This is a news, like in any other city all over the world. What make this article novel should be the specific explanation of Wuhan growth vis à vis different growth paths in other cities, giving originality and novelty to the interpretation of urban growth patterns and processes. Thus Wuhan can be a model for interpreting urban growth not only in China but also in other countries all over the world. Authors should embark in a major revision just to motivate this not secundary aspect. You are publishing on a top level journal and your article should follow the required standard for such target. Thank you.
Author Response

(The authors gave the same response as above.)

Round 2
Reviewer 1 Report
The authors have addressed my comments adequately and I think the article has improved overall.
The only observation that I would like to make concerns the structure of some sentences in the new part of the Policy Recommendations: the sentences from "Under strict land management systems, it is becoming more and more difficult for cities to expand on a large scale, and internal infill urban expansion will become the dominant form." should be rephrased to improve the structure of the text and make concepts clearer.
Author Response
Please see the attachment for more details.

Author Response

(The authors gave the same response as above.)

Reviewer 4 Report
This is a good (standard) paper with a very good revision clarifying some critical points. I agree with publication
Round 3
Reviewer 2 Report
Dear authors,
The paper is looking good now. Just one more suggestion. Please change the phrase:
"so it is clear that the factors affecting the expansion of each city fringe are very complex and each city characterizes different characteristics.”
with
so it is clear that the factors affecting the expansion of each city fringe are very complex and therefore deriving different characteristics.
Best regards,